# A quantitative model reveals a frequency ordering of prediction and prediction-error signals in the human brain

Zenas C. Chao [1✉], Yiyuan Teresa Huang [1,2] & Chien-Te Wu [1,2]

The human brain is proposed to harbor a hierarchical predictive coding neuronal network underlying perception, cognition, and action. In support of this theory, feedforward signals for prediction error have been reported. However, the identification of feedback prediction signals has been elusive due to their causal entanglement with prediction-error signals. Here, we use a quantitative model to decompose these signals in electroencephalography during an auditory task, and identify their spatio-spectral-temporal signatures across two functional hierarchies. Two prediction signals are identified in the period prior to the sensory input: a low-level signal representing the tone-to-tone transition in the high beta frequency band, and a high-level signal for the multi-tone sequence structure in the low beta band. Subsequently, prediction-error signals dependent on the prior predictions are found in the gamma band. Our findings reveal a frequency ordering of prediction signals and their hierarchical interactions with prediction-error signals supporting predictive coding theory.

[1] International Research Center for Neurointelligence (WPI-IRCN), UTIAS, The University of Tokyo, Tokyo, Japan. [2] School of Occupational Therapy, College of Medicine, National Taiwan University, Taipei, Taiwan. ✉email: zenas.c.chao@gmail.com

Predictive coding is an emerging general theory of the functional organization of the brain. The basic principle of this theory is that brain networks continuously generate and update prediction signals representing sensory inputs that, in turn, drive the production of prediction-error signals when the predicted and actual sensory inputs differ[1–4]. According to predictive coding theory, this form of dynamic communication is achieved by a hierarchical and bidirectional cascade of large-scale cortical signaling in order to minimize overall prediction errors. In a highly recursive process, higher-level cortical areas harboring internal models of the world predict inputs from lower-level areas through top-down connections, and prediction-error signals are generated to update the internal models through bottom-up connections.

The predictive coding framework provides broad explanatory power for diverse cognitive processes, such as perceptual decision-making[5–7], expectation-facilitated visual and auditory perception[8,9], and attention[10,11], compared to alternate theories, and has been proposed as a unified model of cognition[12,13]. Predictive coding also offers a plausible neurocomputational mechanism for psychiatric disorders, such as schizophrenia and autism[14–17]. However, direct evidence for some of its core tenets remains lacking, and it is essential to unambiguously identify the theorized prediction and prediction-error signals in brain physiology, and experimentally evaluate their hierarchical flows and interactions.

Prediction-error signals have been extensively studied and commonly characterized as neural responses evoked by unexpected or oddball stimuli. Macroscopic prediction-error signals have been identified as reduced responses to expected stimuli or increased responses to unexpected stimuli in fMRI[18,19], gamma-band oscillations (>40 Hz) in electrocorticography (ECoG)[20–22], and magnetoencephalography (MEG)[23,24], or mismatch negativity responses in EEG[25–27] and MEG[28,29]. At the microscopic level, neuronal responses to unexpected stimuli were observed in layer 2/3 of visual cortex in mice[30], in the central auditory pathway and subcortical regions in rat[31], and in association with increased spiking and gamma-band local field potential (LFP) oscillations in superficial-layer cortex in monkey[32]. The hierarchical organization of prediction-error signals has been examined in a local-global paradigm[25] used to investigate hierarchical auditory processing in human and monkey[10,20,22,27,33,34].

Unlike prediction-error signals, the identification of hierarchical prediction signals has not been achieved because changes in predictions lead to changes in prediction errors and vice versa, and it is difficult using current methods to separate these two interdependent neural processes. As a first step toward understanding prediction signals, a few studies focused on neural responses correlated to sensory predictability. For example, when manipulating the tone sequence from one frequency to another, beta-band (12–30 Hz) activity in ECoG was found to correlate with the change in prediction[35]. Furthermore, beta-band activity in MEG was found to change parametrically with the predictability of action-outcome sequences[23], and enhanced alpha-band and beta-band LFP oscillations were found during predictable stimuli in monkeys[32]. However, manipulating predictability also changes the subsequent prediction-error signal, thus neural responses recorded under these sensory predictabilities contain different prediction errors that cannot be factored out. Another approach is to examine the neural response during omission, based on the argument that it reflects solely the prediction signals since no errors are computed when sensory inputs are absent[27,36]. However, unpredicted omissions also lead to surprises or omission errors, thus omission responses contain both prediction and prediction-error signals.

To disentangle prediction and prediction-error signals, a dynamic causal model has been used to identify top-down functional connectivity that encoded predicted stimuli during a discrimination task when the stimulus predictability was manipulated[7]. In a similar task, a regression model was used to evaluate the latent contributions of predictions and prediction errors in spiking activity[37]. However, how prediction and prediction-error signals interact across functional hierarchies, another fundamental element of predictive coding theory, remains unknown. To identify hierarchical prediction and prediction-error signals, we provided a quantitative definition of these signals based on a mechanistic and hierarchical predictive coding model, where predictions at each hierarchical level are generated to minimize the mean-squared prediction errors received at the same level. This allows us to infer the interactions between prediction and prediction-error signals within and across hierarchies when prediction is manipulated. With this computational strategy, we recorded human EEG data during an auditory local-global paradigm with manipulated stimulus predictabilities at two hierarchies, and used a model-fitting approach to extract prediction and prediction-error signals from the EEG responses by a tensor-based decomposition method[20,38,39], and revealed their spatio-spectro-temporal structures and hierarchical interactions.

Our results provide a comprehensive view of the signal flow and interactions of hierarchical prediction and prediction-error signals in the cortical network. In particular, we show that hierarchical prediction signals are not only spatiotemporally distinctive, but also frequency-specific. More broadly, our combined experimental and analytical approach can be applied to any experimental paradigm where predictability can be defined, and provides a robust platform for the functional mapping of brain-wide predictive coding in normal and disordered brain.

## Results

**Local-global paradigm with manipulated temporal regularities**. Thirty healthy adults were recruited in this study. During the task, participants listened to a series of short tone sequences based on the local-global auditory paradigm while brain activity was recorded by 64-channel EEG. To ensure vigilance, participants were instructed to both visually fixate and attend to the sounds.

Three stimulus items were used to create the short tone sequences: $x$ (standard tone), $y$ (deviant tone), and $o$ (omission, no tone). Each sequence consisted of 2 or 3 stimulus items with one of three temporal structures: (1) the last tone is identical to the preceding tone(s) ($xx$ or $xxx$, jointly denoted as xx), (2) the last tone is different from the preceding tone(s) ($xy$ or $xxy$, jointly denoted as xy), or (3) the last tone is omitted ($xo$ or $xxo$, jointly denoted as xo) (Fig. 1a). Note that $xo$ contained only one stimulus items, but was used to represent an omission in a 2-tone sequence. Similarly, $xxo$ was used to represent an omission in a 3-tone sequence. Sequences were delivered in blocks of 144 trials, which consisted of either only 2-tone sequences ($xx$, $xy$, and $xo$) or only 3-tone sequences ($xxx$, $xxy$, and $xxo$). Eight blocks were used, each with a distinct configuration of the sequence length and trial numbers for xx, xy, and xo (Fig. 1b).

The local-global paradigm is designed to establish predictions with different strengths by varying the degree of temporal regularity at two hierarchical levels. A local regularity is established by the tone transition probability ($TP_x$, $TP_y$, and $TP_o$: the conditional probability of the incoming tone being $x$, $y$, or $o$, respectively, when the previous tone is $x$), which is controlled by the sequence length and sequence ratio. On the other hand, a global regularity is established by a sequence

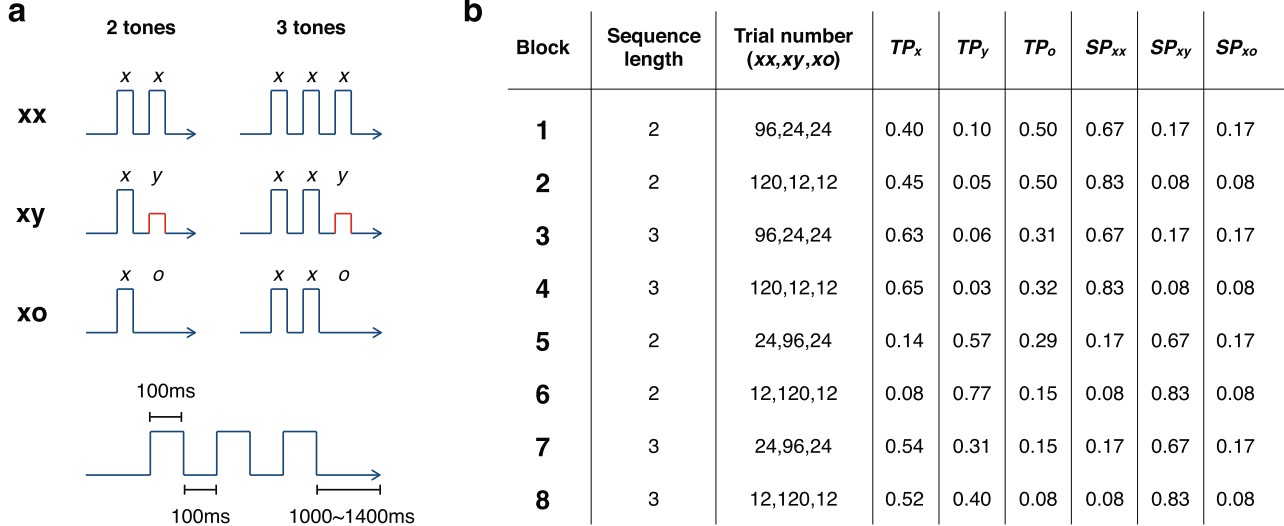

**Fig. 1 Local-global paradigm with manipulated temporal regularities. a** The sequence structures of xx, xy, and xo for the 2-tone and 3-tone sequences. **b** The configurations of the 8 sequence blocks and the corresponding transition and sequence probabilities. The probably values were rounded up to two decimal places.

probability ($SP_{xx}$, $SP_{xy}$, and $SP_{xo}$: the probability of the current sequence being xx, xy, or xo, respectively), which is controlled by the sequence ratio. The transition and sequence probabilities for the 8 blocks are shown in Fig. 1b (see their calculation in "Methods"). To examine the brain responses influenced by these probabilities, we eliminated tone-specific effects by delivering each block twice (one run with a low-pitched tone $A$ as $x$ and a high-pitched $B$ as $y$, and the other run with tone $B$ as $x$ and tone $A$ as $y$), and merged the EEG data from two runs for analysis (see "Methods").

**A hierarchical predictive coding model for the local-global paradigm.** We instantiated a hierarchical predictive coding model to extract the underlying signals and their interdependence during the local-global paradigm. This allowed us to decompose the EEG data based on quantitative model predictions and identify the neural responses for each prediction and prediction-error component at each level of the hierarchy. The model consists of three hierarchical levels (Level S, Level 1, and Level 2) and two streams (x stream and y stream). Level S is the sensory level that receives thalamic input, which was a value between 0 and 1, Level 1 learns and encodes the local regularity (transition probabilities), and Level 2 learns and encodes the global regularity (sequence probabilities). The x and y streams process the tone $x$ and $y$, respectively. Importantly, the model focuses on the interactions between prediction and prediction-error signals during the last tone of a sequence after both local and global regularities are learned.

Figure 2a shows the neural operations in the x stream between Levels S and 1. Level S contains a neuronal population (denoted by $x_s$) that receives a sensory input (black arrow) and a prediction signal (green arrow) from Level 1, and sends a prediction-error signal (blue arrow) to Level 1. Level 1 contains a neuronal population ($x_1$) that receives the prediction-error signal from Level S, and sends a prediction signal to Level S. If we assume that the strengths of the sensory input and the prediction signal are 1 and $P1_x$ ($0 \leq P1_x \leq 1$), respectively, then there are two possible situations: (1) if the last tone is $x$, then the strength of the prediction-error signal is $1 - P1_x$, (2) if the last tone is not $x$ (either the tone is $y$ or omitted), then the prediction error is $0 - P1_x$ (a negative value), and the strength of the corresponding prediction-error signal is $|0 - P1_x| = P1_x$ ($|\cdot|$ indicates the absolute value).

Absolute values are taken because we assume predictions and prediction errors are encoded in neuronal firing rates, a most straightforward scheme for encoding probabilistic representations and computations[40], which can only have non-negative values. Thus, the prediction-error signal received at Level 1 in the x stream during the last tone (denoted as $PE1_x$) is either $1 - P1_x$ or $P1_x$, where the probability of receiving the former is the transition probability from tone $x$ to $x$ ($TP_x$) and the probability of receiving the latter is $1 - TP_x$ (see the bar graph in Fig. 2a).

Figure 2b shows the neural operations in the x stream between Levels 1 and 2. Similar to Level 1, Level 2 contains a neuronal population ($x_2$) that receives the prediction-error signal from Level 1, and sends a prediction signal $P2_x$ to Level 1. If the sequence is xx, then the prediction-error signal received at Level 1 is $1 - P1_x$ (since Level S receives tone $x$) and the prediction-error signal received at Level 2 is $|1 - P1_x - P2_x|$. If the sequence is not xx (xy or xo), then the prediction-error signal received at Level 1 is $P1_x$ (since Level S receives not $x$) and the prediction-error signal received at Level 2 is $|P1_x - P2_x|$. Thus, the prediction-error signal received at Level 2 in the x stream during the last tone (denoted as $PE2_x$) is either $|1 - P1_x - P2_x|$ or $|P1_x - P2_x|$, where the probability of receiving the former is the sequence probability of sequence xx ($SP_{xx}$) and the probability of receiving the latter is $1 - SP_{xx}$.

Figure 2c shows the complete model during the last tone in xx, xy, and xo sequences. We assume that the strengths of prediction signals ($P1_x$, $P2_x$, $P1_y$, and $P2_y$) reach steady-state values when the transition and sequence probabilities in a given block are learned. We note that the same prediction signals appear in all sequences (xx, xy, and xo), since predictions occur before the last tone arrives. Furthermore, even though the $x$ and $y$ tones are processed in separate streams based on the tonotopic organization, two streams need to integrate information at Levels 1 and 2 to compute transition probabilities ($TP_x$, $TP_y$, and $TP_o$) and sequence probabilities ($SP_{xx}$, $SP_{xy}$, and $SP_{xo}$), respectively. In Fig. 2c, we indicate these integrations for probability computations as horizontal gray bars between populations $x_1$ and $y_1$ and between populations $x_2$ and $y_2$.

We further added some tunings to the model across different levels (see Fig. 2d). At Level S, a scaling factor $s_0$ was added to the sensory input in the x stream to account for the sensory adaption for the repetitive tone $x$. The value of $s_0$ was between 0 and 1,

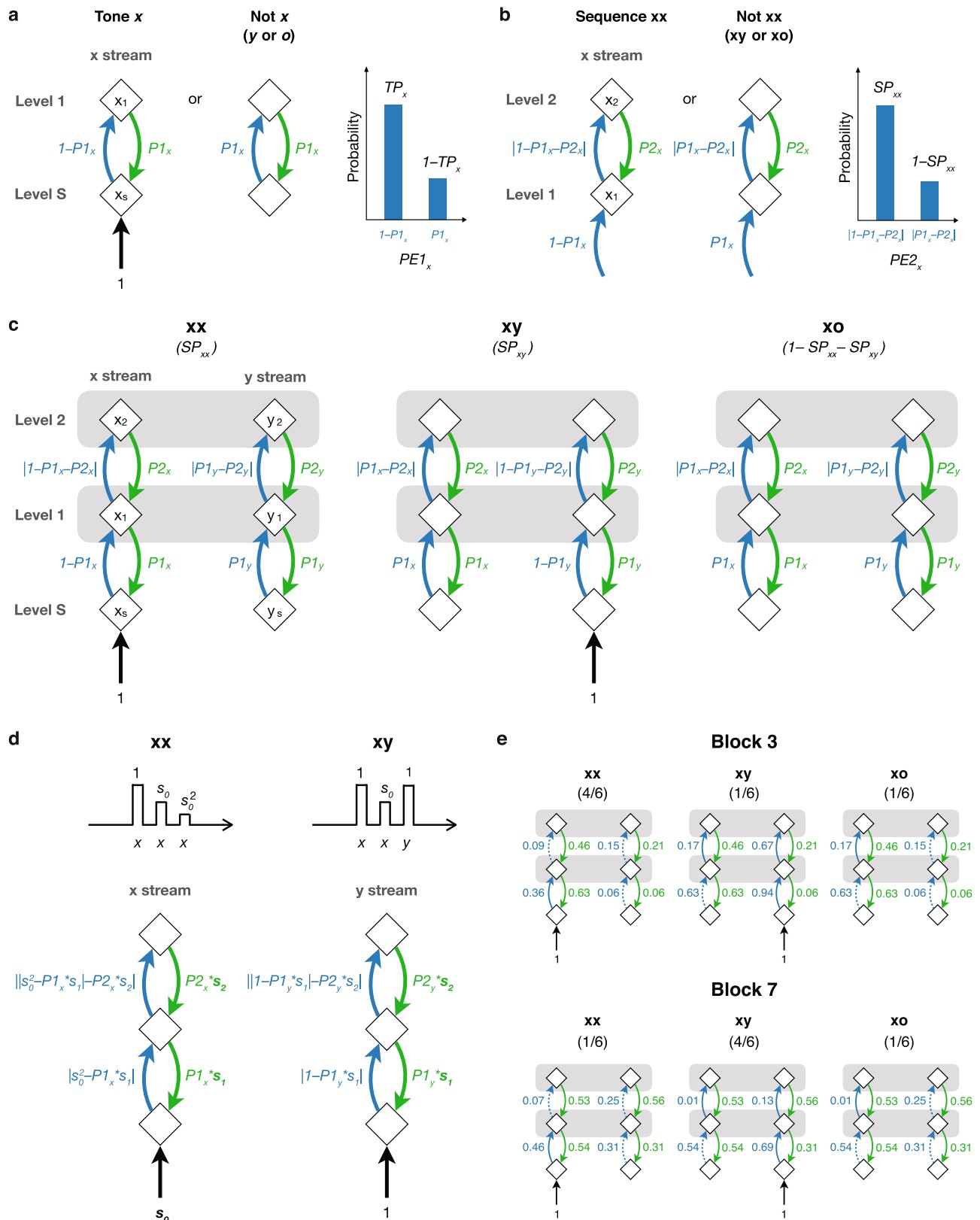

where $s_0 = 1$ represents no sensory adaptation. Therefore, in the $n$-tone xx sequence where tone $x$ is repeated $n-1$ times, the PE1 and PE2 in the $x$ stream are $|s_0^{n-1} - P1_x|$ and $|s_0^{n-1} - P1_x - P2_x|$, respectively. For the xy sequence, since tone $y$ does not repeat, adaption does not occur in the $y$ stream. At Levels 1 and 2, we added scaling factors $s_1$ and $s_2$ to the first-level predictions ($P1_x$

and $P1_y$) and the second-level predictions ($P2_x$ and $P2_y$), respectively, to account for imperfect predictions. When $s_1 = 1$ and $s_2 = 1$, the predictions are optimal (see how the optimal predictions were quantified below). When $s_1 < 1$ or $s_2 < 1$, the predictions are hypo-sensitive to the inputs. For example, if $s_1 = 0$, there will be no first-level prediction. When $s_1 > 1$ or $s_2 > 1$,

**Fig. 2 A hierarchical predictive coding model for the local-global paradigm. a** The proposed neural operations in the x stream between Levels S and 1 during the presentation of tone x or not. An explanatory illustration of the probability distribution of the first-level prediction error in the x stream ($PE1_x$) is shown on the right. The neuronal populations (diamonds with labels $x_s$ or $x_1$), prediction-error signal (blue arrow), prediction signal (green arrow), and sensory input (black arrow) are shown. **b** The neural operations in the x stream between Levels 1 and 2. **c** The complete model during the last tone in xx, xy, and xo sequences. The horizontal gray bars at Levels 1 and 2 indicate integration between the x and y streams for computing transition and sequence probabilities, respectively. **d** Model tunings with $s_0$, $s_1$, and $s_2$. A decreased response (scaled by $s_0$) to repeated tone x during the xx sequence, and a decreased response to repeated tone x with a fresh response to tone y during the xy sequence. The corresponding models of the last tone are also shown, where P1 and P2 are scaled by $s_1$, and $s_2$, respectively. **e** An example of the strengths of prediction and prediction-error signals in Blocks 3 and 7 with no tunings ($s_0 = s_1 = s_2 = 1$). The negative errors, where the prediction is greater than the input or prediction error to be predicted, are shown in blue dashed arrows.

the predictions are hyper-sensitive to the inputs, where the corresponding transition or sequence probabilities are over-estimated. Note that $s_1$ and $s_2$ were applied to both the x and y streams, since erroneous estimation of transition or sequence probabilities could occur at both streams.

Next, we asked what steady-state values the prediction signals ($P1_x$, $P2_x$, $P1_y$, and $P2_y$) will reach when the transition and sequence probabilities are learned. We propose a simple model where the optimal value of each prediction signal is to minimize the mean-squared error received (see model calculation in "Methods"). Based on the model, all prediction signals are determined once the transition probabilities ($TP_x$ and $TP_y$), sequence probabilities ($SP_{xx}$ and $SP_{xy}$), and scaling factors ($s_o$, $s_1$, and $s_2$) are known. The transition probabilities can be calculated based on the number of tones in a sequence ($n$) and the sequence probabilities. It is important to note that the transition from tone x to o occurs not only during the xo sequence, but also at the end of the xx sequence where the last x tone is followed by no tone. Since Level 1 simply predicts what will happen after an x tone and makes no distinction between the two cases, the x to o transitions in both the xo and xx sequences were considered in the transition probability calculation (see "Methods"). Examples of the strengths of the prediction and prediction-error signals in Blocks 3 and 7 with $s_0 = s_1 = s_2 = 1$ (the optimal predictions with no sensory adaptation) are shown in Fig. 2e.

**Model prediction: prediction and prediction-error components in contrast responses**. The model identifies the prediction and prediction-error signals present in each trial type during each of the 8 blocks, and predicts how much these signals remain when we contrast different trial types. These model predictions form the basis of our data-fitting decomposition analysis to extract prediction and prediction-error components from the EEG signals. To achieve this, we used the model predictions in 16 within-block contrasts and 24 across-block contrasts (Fig. 3a). Within each block, there were three possible contrasts: between the xy to xx sequences (xy – xx), between the xo to xx sequences (xo – xx), and between the xy to xo sequences (xy – xo). Since xy – xo is equivalent to the difference between xy – xx and xo – xx, i.e., (xy – xx) – (xo – xx), it was excluded from the analysis. Therefore, among 24 possible within-block contrasts (3 contrasts per block × 8 blocks), only 16 were included in the analysis (shown as blue arrows in Fig. 3a).

Conversely, the across-block contrasts compare the same sequence from two blocks. For a 2-tone sequence (xx, xy, or xo), there were 6 possible across-block contrasts: Blocks 1 – 2, 2 – 6, 6 – 5, 5 – 1, 1 – 6, and 2 – 5. Since Blocks 2 – 5 is equivalent to the combination of Blocks 5 – 1 and 1 – 2, i.e., (5 – 1) – (1 – 2) it was excluded from the analysis. Similarly, Blocks 1 – 6 is equivalent to the combination of Blocks 1 – 2 and 2 – 6, and was excluded from the analysis. Therefore, among 18 possible across-block contrasts for 2-tone sequences (6 contrasts per sequence × 3 sequences), 12 were included in the analysis.

By adding another 12 contrasts for 3-tone sequences, a total of 24 across-block contrasts were included in the analysis (shown as green arrows in Fig. 3a).

In the 16 within-block contrasts, only prediction-error signals remain since the prediction signals are the same within each block. Figure 3b shows the model predictions of the remaining prediction-error signals at the first level ($PE1 = PE1_x + PE1_y$) and the second level ($PE2 = PE2_x + PE2_y$) when $s_0 = s_1 = s_2 = 1$. Note that while both PE1 and PE2 are positive values, their contrast values between two trial types can be negative. On the other hand, in the 24 across-block contrasts, both prediction and prediction-error signals remain. The model predictions of the remaining prediction signals at the first level ($P1 = P1_x + P1_y$) and the second level ($P2 = P2_x + P2_y$) and prediction-error signals (PE1 and PE2) are shown in Fig. 3c (only the model with both positive and negative errors is shown for clarity). Note that we assume that EEG recordings offer insufficient spatial resolution to separate the x and y streams, therefore, the model predictions focus on P1, P2, PE1, and PE2, where the x and y streams are combined.

**Model-fitting: optimal decomposition of EEG data**. Our strategy to extract prediction and prediction-error signals from EEG data was to perform the within-block and across-block contrasts on EEG signals and factorize the contrast responses into components predicted by the model. In other words, EEG components of P1, P2, PE1, and PE2 should have distinct and unique contributions to contrast responses in the 16 within-block and 24 across-block contrasts, as shown in Fig. 3b, c.

To demonstrate this strategy, here we use the within-block contrasts as an example. Figure 4a shows an example of the contrast response at a single channel (channel 22) in a single within-block contrast (xy – xx in Block 3, or contrast 3 in Fig. 3a). Here, the contrast response was quantified as the significant difference in event-related spectral perturbation (ERSP) of the current source density (CSD) between the xy and xx sequences across subjects ($p < 0.05$, 30 subjects, 1000 bootstrapping, two-sided, false discovery rate correction) (see "Methods"). Figure 4b shows the contrast responses from the same channel in all 16 within-block contrasts, where distinct patterns were observed around the last tone (time zero). Please see the overall occurrence of significant contrast responses across all channels for both the within-block and across-block contrasts in Supplementary Fig. 1.

To evaluate the model, we acquired a more comprehensive view of the contrast responses across the multi-dimensional space of channels, time, frequencies, and contrasts. This classification was achieved using an unbiased decomposition analysis that extracts latent components hidden within functional network dynamics[20,38,39] (see "Methods"). We first pooled significant contrast responses from all channels and all contrasts to create a broadband library. To organize and visualize this dataset, we created a tensor with three dimensions: *Channel* (brain area), *Time-Frequency* (in-trial dynamics), and *Contrast* (contrast

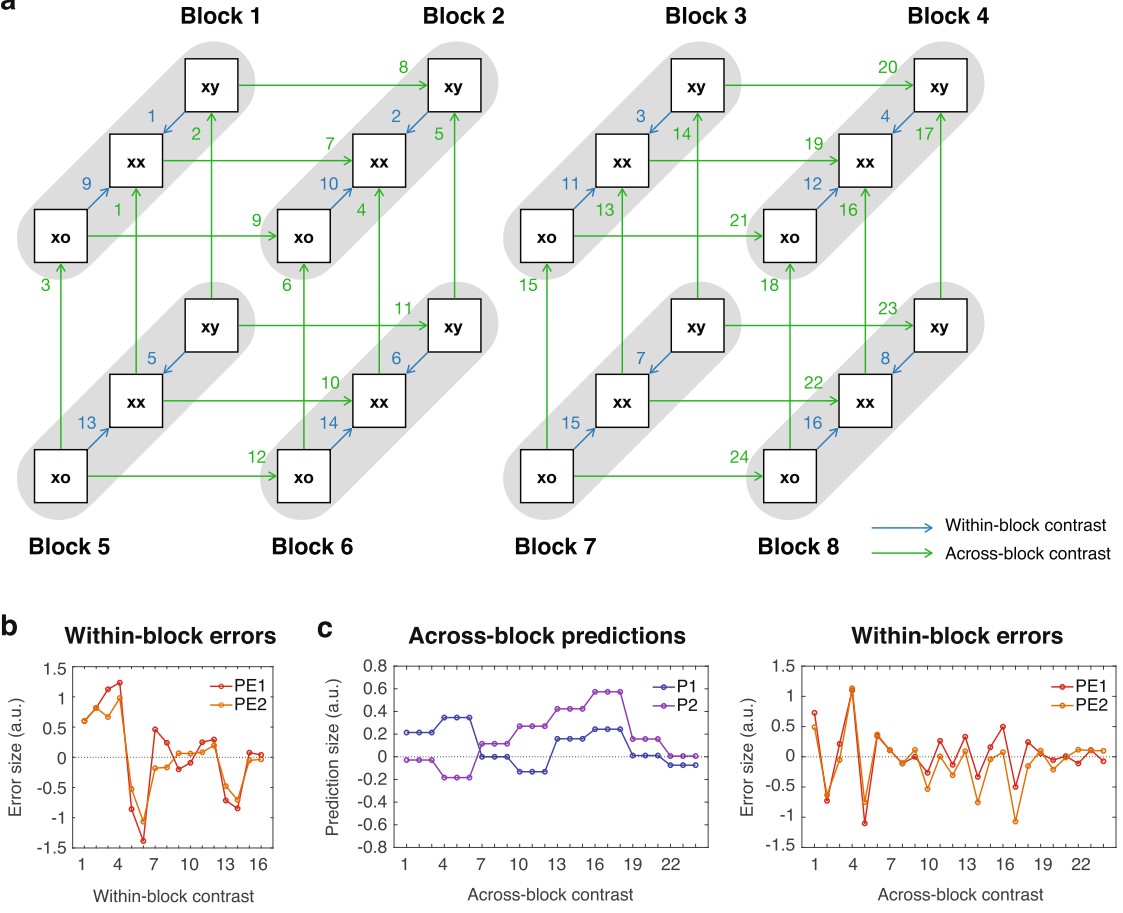

**Fig. 3 Block comparisons and model predictions. a** The 16 within-block contrasts and the 24 across-block contrasts. Each gray bar represents a block (labeled), where each block contains 3 types of trials (xy, xx, and xo). The within-block and across-block contrasts are indicated by blue and green arrows, respectively. The contrast indices are also shown. **b** The model prediction of prediction-error signals in the 16 within-block contrasts. The indices of contrasts (1 to 16) are indicated in panel **a**. Only prediction-error signals are shown, since no prediction signals are left after the within-block contrasts. **c** The model prediction of prediction signals and prediction-error signals in the 24 across-block contrasts. The indices of contrasts (1 to 24) are indicated in panel **a**.

response), for the anatomical, dynamic, and functional aspects of the data, respectively. The dimensionality of the tensor was 60 (channels) by 37,500 (375 time points and 100 frequency bins) by 16 (within-block contrasts) or 24 (across-block contrasts). To extract structured information from the dataset, we factorized the 3D tensor into multiple components by performing parallel factor analysis (PARAFAC), a generalization of principal component analysis (PCA) to higher-order arrays[41], and measured the consistency of factorization under different numbers of components[42] (see "Methods").

For the within-block contrasts, we first performed a model-free data-driven decomposition to examine how many structured components were contained in the contrast responses. A consistency of 100% was obtained when the tensor was factorized into two components, while it dropped significantly when the tensor was factorized into three components (Fig. 4c). This indicated that there were two consistent components in the pooled contrast responses, where each component contained a unique fingerprint of network anatomy, dynamics, and function (see these components in Supplementary Fig. 2). To further test whether the two dominant components were associated with PE1 and PE2, we repeated the factorization with the third dimension *Contrast* fixed with the values proposed by the model (the 16 values in Fig. 3b). This model-driven analysis was performed by using models with different scaling factors $s_0$ (between 0 and 1), $s_1$

(between 0 and 2), and $s_2$ (between 0 and 2). The best-fitting model with the smallest residual sum of squares (RSS) was found with a consistency of 90% when $s_0 = 0.3$, $s_1 = 1.2$, and $s_2 = 1.0$ (Fig. 4d). This suggested that the total contrast responses consisted of two structured components that subserved PE1 and PE2.

For the across-block contrasts, the model-free data-driven analysis indicated three consistent components in the pooled contrast responses, where consistency dropped from 85 to 35% when factorizing the tensor from 3 to 4 components (Fig. 4e). We hypothesized that the 3 components in the data represented P1, P2, and the combination of PE1 and PE2, since the model predictions of P1 and P2 showed distinct patterns while the model predictions of PE1 and PE2 are highly correlated and thus difficult to be separated (see Fig. 3c). Therefore, for the model-driven analysis, we factorized the total contrast responses with the third dimension fixed with the predicted values of P1, P2, and $a*PE1 + (1-a)*PE2$ from the model, where $a$ was a weighting factor between 0 and 1. The best-fitting model with the smallest RSS was found with a consistency of 85% when $s_0 = 0.3$, $s_1 = 0.8$, $s_2 = 1.0$ (Fig. 4f), and when $a = 0.5$ (see Supplementary Fig. 3). This suggested that the identified components subserved P1, P2, and overall prediction errors (PE1 + PE2).

To explore further, we considered two distinct computations for prediction-error signals: a positive-error computation when

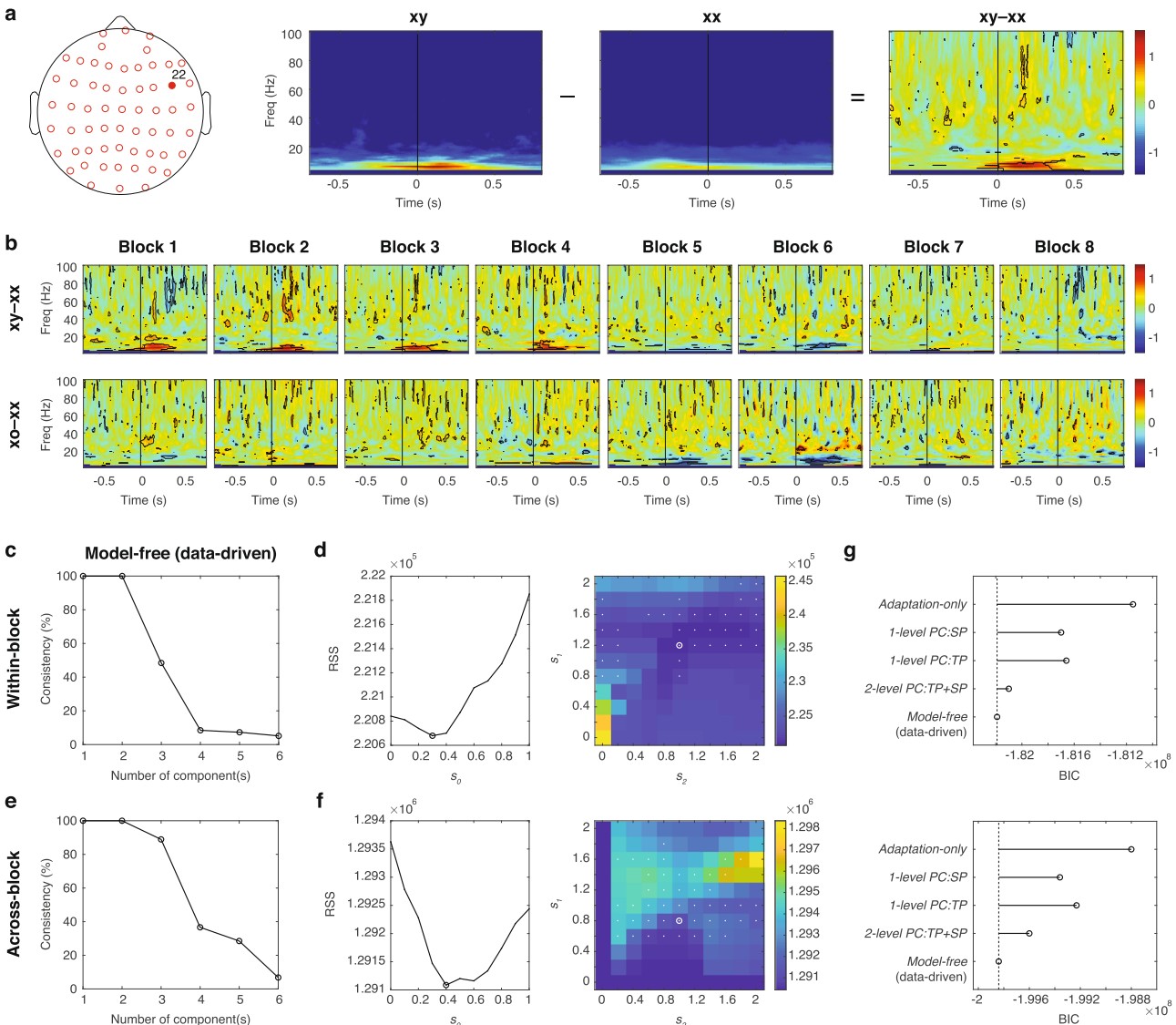

**Fig. 4 Data-fitting and model comparisons. a** An example of the contrast response at channel 22 (location shown on a head map) in a within-block contrast (xy – xx in Block 3). The average ERSPs for the xy and xx sequences and their contrast response (xy – xx) are shown. The significant difference is indicated by the black contour. The time zero represents the onset of the last tone. **b** The contrast responses from the same channel in all 16 within-block contrasts. **c** The consistencies of data-driven factorization for contrast responses from the within-block contrasts. **d** The optimal parameters for data-fitting for the within-block contrasts. Only models with a fitting consistency >80% were considered. For each $s_0$, the minimal residual sum of squares (RSS) across different combinations of $s_1$ and $s_2$ is shown in the left panel. The minimal RSS was found when $s_0 = 0.3$ (indicated by a black circle). The combination of $s_1$ and $s_2$ under this minimum is indicated by a white circle in the right panel. Models with a fitting consistency >80% are indicated by white dots. The color bar represents RSS. **e** The consistencies of data-driven factorization for the across-block contrasts. **f** The optimal parameters for data-fitting for the across-block contrasts. The same representation is used as in panel **d**. **g** Model comparisons for the within-block and across-block contrasts. The vertical dash line indicates the benchmark from the model-free data-driven decomposition.

the prediction is smaller than the sensory input or prediction error, and a negative-error computation when the prediction is greater than the sensory input or prediction error (see detailed discussion on positive and negative prediction errors in Discussion). Since different error computations could involve different neural mechanisms or occur at different cortical layer, which could differentially contribute to the EEG signal, we tested models with different error computations, and the best-fitting model was found when both positive- and negative-error computations were considered (see Supplementary Fig. 4). We also tested models with different transition probability calculations: (1) a model considered only the overall occurrences of tones and the transitions between them were neglected (denoted by *No-transition*), and (2) a model neglected transitions from

tone x to o at the end of the xx sequence (denoted by *No-ending*). Compared to the proposed model, these two alternative models showed higher RSS (see Supplementary Fig. 5), which suggested that lower-level predictions manage both tone transitions and sequence endings.

In summary, our proposed model fitted the data with high consistencies, and the best-fitting models indicated that (1) sensory adaptation was needed to explain the data ($s_0 \approx 0.3$, which was consistent with the adaptation factor measured directly from EEG responses shown in Supplementary Fig. 6), (2) predictions at both levels were close to optimal ($s_1 \approx s_2 = 1.0$), and (3) prediction-error signals in EEG responses could represent the combination of positive- and negative-error computations. To visualize the prediction and prediction-error components, we

**Table 1 Model comparisons.**

| Model | Optimal parameters | Number of data point ($u$) | Number of estimated elements ($w$) | Residual sum of squares (RSS) | Bayesian information criterion (BIC) |
|---|---|---|---|---|---|
| Within-block contrasts | | | | | |
| Model-free | N/A | $60 \times 37{,}500 \times 16$ | $(60 + 37{,}500 + 16) \times 2$ | $2.2013e + 05$ (Consistency = 100%) | $-1.8219e + 08$ |
| 2-level PC:TP + SP | $s_0 = 0.3$ $s_1 = 1.2$ $s_2 = 1.0$ | $60 \times 37{,}500 \times 16$ | $(60 + 37{,}500) \times 2$ | $2.2068e + 05$ (Consistency = 90%) | $-1.8210e + 08$ |
| 1-level PC:TP | $s_0 = 0.2$ $s_1 = 1.4$ | $60 \times 37{,}500 \times 16$ | $(60 + 37{,}500) \times 1$ | $2.2747e + 05$ | $-1.8166e + 08$ |
| 1-level PC:SP | $s_0 = 0.2$ $s_1 = 0.6$ | $60 \times 37{,}500 \times 16$ | $(60 + 37{,}500) \times 1$ | $2.2720e + 05$ | $-1.8170e + 08$ |
| Adaptation-only | $s_0 = 0.3$ $\tau_0 = 1.3$ s | $60 \times 37{,}500 \times 16$ | $(60 + 37{,}500) \times 1$ | $2.3071e + 05$ | $-1.8115e + 08$ |
| Across-block contrasts | | | | | |
| Model-free | N/A | $60 \times 37{,}500 \times 24$ | $(60 + 37{,}500 + 24) \times 3$ | $1.2854e + 06$ (Consistency = 99%) | $-1.9984e + 08$ |
| 2-level PC:TP + SP | $s_0 = 0.4$ $s_1 = 0.8$ $s_2 = 1.0$ | $60 \times 37{,}500 \times 24$ | $(60 + 37{,}500) \times 3$ | $1.2911e + 06$ (Consistency = 85%) | $-1.9960e + 08$ |
| 1-level PC:TP | $s_0 = 0.4$ $s_1 = 2.0$ | $60 \times 37{,}500 \times 24$ | $(60 + 37{,}500) \times 2$ | $1.3163e + 06$ | $-1.9923e + 08$ |
| 1-level PC:SP | $s_0 = 0.4$ $s_1 = 1.0$ | $60 \times 37{,}500 \times 24$ | $(60 + 37{,}500) \times 2$ | $1.3131e + 06$ | $-1.9936e + 08$ |
| Adaptation-only | $s_0 = 0.3$ $\tau_0 = 1.5$ s | $60 \times 37{,}500 \times 24$ | $(60 + 37{,}500) \times 1$ | $1.3433e + 06$ | $-1.9880e + 08$ |

later used the same parameters for the within-block and across-block contrasts: $s_0 = 0.3$, $s_1 = s_2 = 1.0$.

**Model comparison: alternative predictive coding and adaptation-only models**. To further validate the model, we introduced alternative models with different architectures and mechanics to fit the same EEG data and compare their performance. In addition to the proposed model, a two-level predictive coding model based on both transition and sequence probabilities (denoted by *2-level PC:TP + SP*), we tested: (1) a single-level predictive coding model with only transition probability (*1-level PC:TP*), (2) a single-level predictive coding model with only sequence probability (*1-level PC:SP*), and (3) an adaptation-only model with no predictive coding mechanisms (*Adaptation-only*). We also used the model-free data-driven decomposition (*Model-free*, as shown in Fig. 4c, e), which provided the optimal and unbiased description of the data, as a benchmark.

The single-level predictive coding models were similar to the one shown in Fig. 2, but with only Levels S and 1. For *1-level PC:TP* and *1-level PC:SP*, P1 at Level 1 minimized the mean squares of PE1 based on transition and sequence probabilities, respectively. Also, a scaling factor $s_0$ was added to the sensory input and a scaling factor $s_1$ was added to P1 (similar to the proposed model in Fig. 2d). The optimal parameters ($s_0$ and $s_1$) for *1-level PC:TP* and *1-level PC:SP* for the within-block and across-block contrasts are shown in Table 1 and Supplementary Fig. 7.

For *Adaptation-only*, sensory adaptation was modeled by two parameters: a scaling factor $s_0$ (between 0 and 1) and a time constant $\tau_0$ (between 0.1 and 2 s, including timescales from a single tone to multiple sequences). $s_0$ determines the maximal response that can be evoked immediately after receiving a stimulus. If $s_0 = 1$, there is no adaptation, and the next stimulus can immediately evoke a full response. On the other end, if $s_0 = 0$, the next stimulus cannot evoke any response immediately after the previous stimulus, however, this effect can recover over time

with a time constant $\tau_0$. The model describes the trial-by-trial responses during each block, and thus predicts the contrast responses in the within-block and across-block contrasts (see example and details in Supplementary Fig. 8). We then fitted the EEG data with these predicted values, and the optimal parameters were found to be: $s_0 = 0.3$ and $\tau_0 \approx 1.4$ s (see Table 1 and Supplementary Fig. 8). This indicated that in *Adaptation-only*, adaption with a timescale that covered multiple sequences was needed.

For model comparison, we quantified the goodness of fit by using the Bayesian information criterion (BIC), which penalizes models with more variables (see details in Table 1). For both the within- and across-block contrasts, our proposed model showed the best fitting with the BIC significantly lower than the alternative models (the between-model differences in BIC were greater than 10, which corresponded to a 150:1 odds that the proposed model was the better fitting model[43]), and was close to the model-free data-driven results (Fig. 4g, more details in Table 1). This indicated that the proposed hierarchical cascade of prediction and prediction-error signals was most suitable to describe the neural processes in the observed data.

**Prediction-error signals extracted from within-block contrasts**. Here we visualized the two components obtained from the within-block contrasts based on the proposed model (with $s_0 = 0.3$, $s_1 = s_2 = 1.0$). These components were visualized by their composition in the three tensor dimensions (Fig. 5a–c). The first component was PE1, which appeared in centrocephalic areas (C3, C4) (Fig. 5a), immediately after the last tone in both lower-frequency (<20 Hz) and higher-frequency (>40 Hz) bands (Fig. 5b). The contributions of this spatio-spectro-temporal pattern to the contrast responses across the 16 within-block contrasts were fixed thus identical to PE1 in the model prediction (Fig. 5c). Note that Fig. 5c differs slightly from the model predictions in Fig. 3b (compare to the model of POS + NEG) due to different adaptation factors were used ($s_0 = 0.3$ and 1, respectively). The

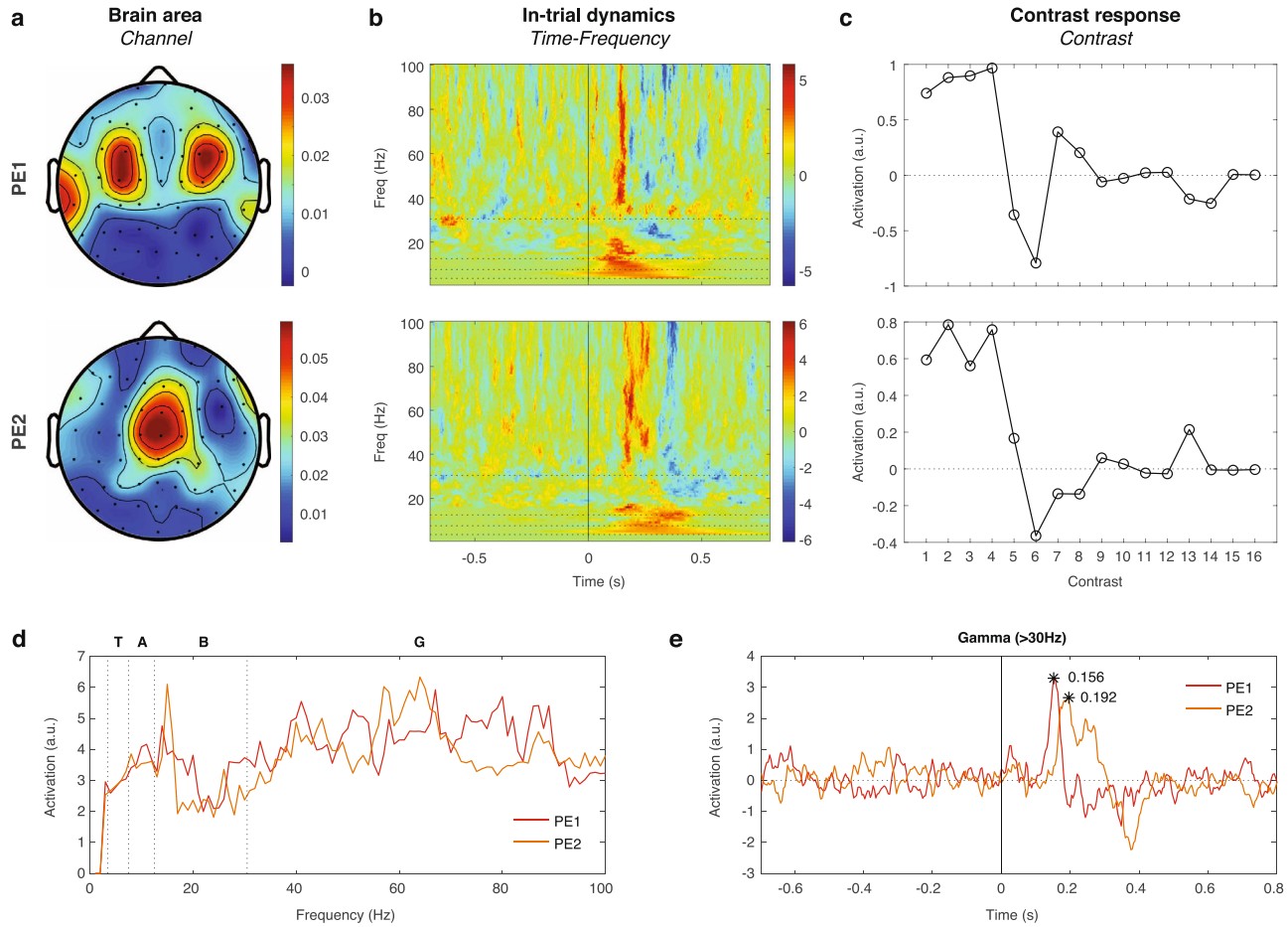

**Fig. 5 Neural signatures of PE1 and PE2 extracted from within-block contrasts. a** The *Channel* dimension of the PE1 and PE2 components extracted from the optimal model. **b** The *Time-Frequency* dimension of the PE1 and PE2 components extracted from the optimal model. The time-frequency representation was normalized to equalize the visualization across frequencies, where each value was divided by the standard deviation within the corresponding frequency. The horizontal dashed lines are drawn at 4, 8, 13, and 30 Hz to indicate different frequency bands in panel **d**. **c** The contributions of PE1 and PE2 to the contrast responses in the 16 within-block contrasts. The values were based on the optimal model prediction and fixed in the model-driven factorization. **d** The spectral profiles of PE1 and PE2. The frequency bands are labeled on the top (T: theta, A: alpha, B: beta, and G: gamma). **e** The temporal profiles of PE1 and PE2 in the gamma band. The peak values are labeled with black asterisks to show the timings of maximal activations.

second component was PE2, which appeared in the central midline area (Cz) (Fig. 5a), slightly after PE1 in both lower- and higher-frequency bands (Fig. 5b), and with the contribution profile of PE2 in the optimal model prediction (Fig. 5c).

Spatially, PE1 represented a source of bilateral auditory cortices, as evidenced by similar CSD-based distribution linked to auditory processing in other human studies[44–46]. On the other hand, PE2 distribution represented a source of frontal cortex, as evidenced by similar CSD-based distribution linked to the medial prefrontal cortex or dorsal anterior cingulate cortex[47–49]. To examine the spectral profile of the PE1 and PE2 components, we measured their maximal activation at each frequency bin in the *Time-Frequency* dimension, and showed that PE1 and PE2 were strongest in the gamma frequency band (31–100 Hz) (Fig. 5d). We further averaged their *Time-Frequency* dimension across the gamma band, and showed that PE1 and PE2 peaked at 156 ms and 192 ms after the last tone, respectively (Fig. 5e). The temporal dynamics of PE1 and PE2 in different frequency bands: theta (4–7 Hz), alpha (8–12 Hz), beta (13–30 Hz), and gamma (31–100 Hz) are shown in Supplementary Fig. 9.

**Prediction signals extracted from across-block contrasts.** For the across-block contrasts, the three components obtained from the proposed model (with $s_0 = 0.3$, $s_1 = s_2 = 1.0$) were visualized

by their compositions in the three tensor dimensions (Fig. 6a–c). The first component was P1, which appeared in the central midline area (Cz) (Fig. 6a), slightly before the last tone in the beta band (Fig. 6b), and with the contribution profile of P1 in the optimal model prediction (Fig. 6c). The second component was P2, which appeared in the frontal and the frontocentral regions (Fig. 6a), slightly before the last tone in the beta band (Fig. 6b), and with the contribution profile of P2 in the optimal model prediction (Fig. 6c). The third component was PE1 + PE2, which appeared in centrocephalic (C3, C4) and central midline (Cz) areas (Fig. 6a), after the last tone in both lower-frequency (<20 Hz) and higher-frequency (>40 Hz) bands (Fig. 6b), and with the contribution profile identical to PE1 + PE2 in the model prediction (Fig. 6c). Note that this component was comparable to the sum of PE1 and PE2 identified previously (see Fig. 5a, b), where their correlations were 0.96 and 0.71 (Pearson's correlation coefficient) in the first dimension (*Channel*) and the second dimension (*Time-Frequency*), respectively (see Supplementary Fig. 10).

To examine the spectral profile of the P1 and P2 components, we measured their maximal activation at each frequency bin in the *Time-Frequency* dimension, and showed that P1 and P2 were both strongest in the beta band but P1 peaked around 23 Hz while P2 peaked around 15 Hz (Fig. 6d). This indicated that

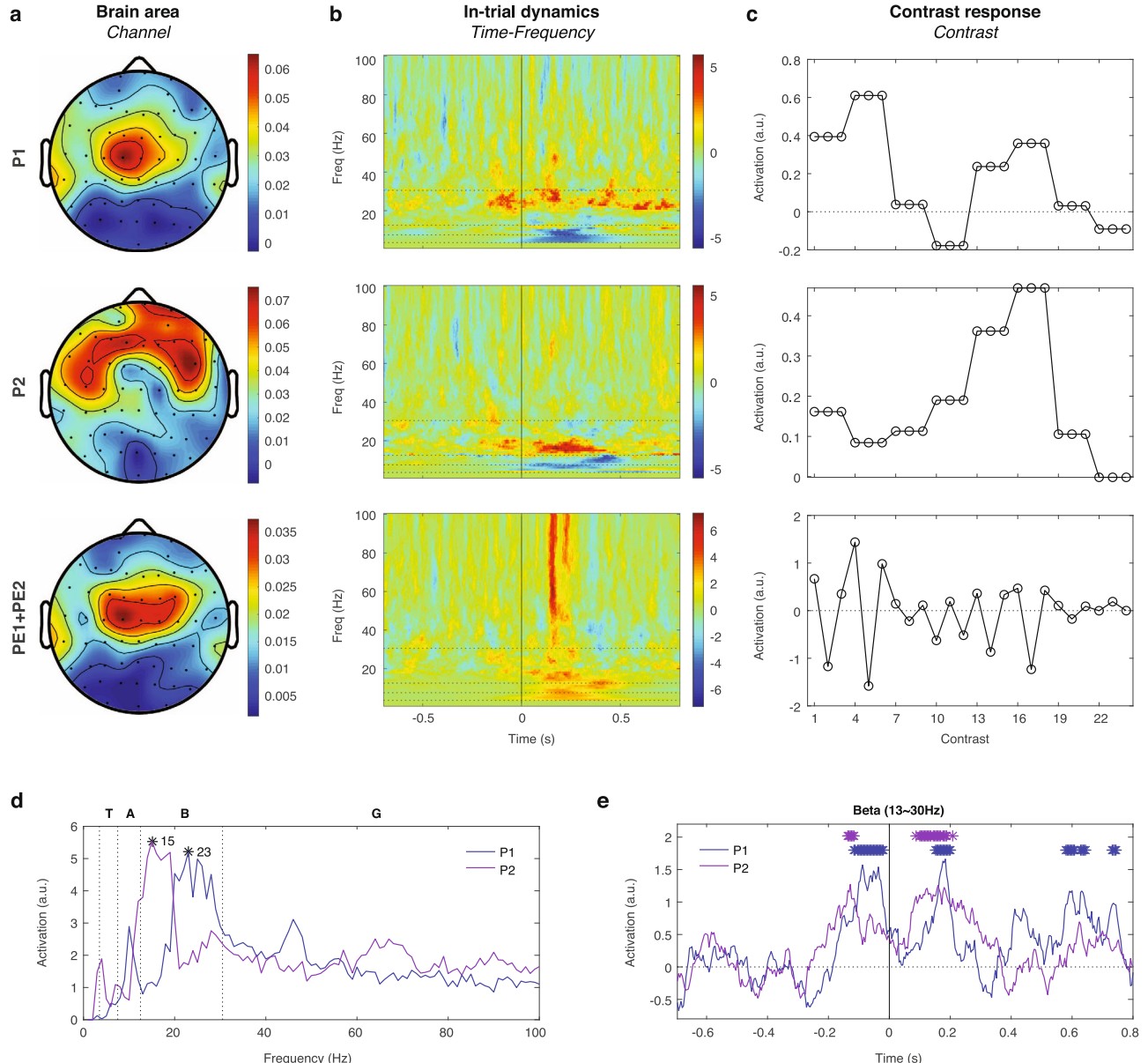

**Fig. 6 Neural signatures of P1 and P2 extracted from across-block contrasts.** The three components obtained from the model-driven factorization on the across-block contrast responses. **a** The *Channel* dimension of the P1, P2, and PE1&PE2 components extracted from the optimal model. **b** The *Time-Frequency* dimension of the extracted components. The same presentation is used as in Fig. 5b. **c** The contributions of the extracted components to the contrast responses in the 24 across-block contrasts. The values were based on the optimal model prediction and fixed in the model-driven factorization. **d** The spectral profiles of P1 and P2. The frequency bands are labeled on the top as in Fig. 5d. The peak values are labeled with black asterisks to show the frequencies of maximal activations. **e** The temporal profiles of P1 and P2 in the beta band. The values significantly greater than 3 times of the standard deviation of values during the baseline period (from −0.7 s to −0.4 s) are shown as asterisks with the corresponding colors.

predictions were frequency-specific, and suggested that predictions of faster events (e.g., transitions between tones) were encoded in faster neural oscillation frequency bands, and predictions of slower events (e.g., sequence occurrences) were encoded in slower neural oscillations. We further averaged the *Time-Frequency* dimension across the beta band, and defined significant activation as a value 3 times greater than the standard deviation of the corresponding baseline values during −0.7 s ~ −0.4 s. The results showed that P1 and P2 were activated before the last tone (Fig. 6e), which indicated that predictions of the last tone were activated prior to its onset, suggesting a proactive and preparatory effect. Furthermore, P1 and P2 were activated again after the last tone, and we theorize that the former was triggered by the last tone to predict the next incoming tone

that was absent, and the latter canceled this omission error to predict the sequence ending (see discussion on signal flows in the "Discussion" section).

**Prediction-error signaling during the learning phase**. Our model assumed that the transition and sequence probabilities were learned. Our next step was to examine the learning process by evaluating changes in the within-block contrast responses from the early phase (first half of the trials) to the late phase (second half of the trials). Based on the model, the P1 and P2 signals were canceled out from the within-block contrast, thus the learning effect was evaluated by changes in the PE1 and PE2 components in each phase. The overall within-block contrast

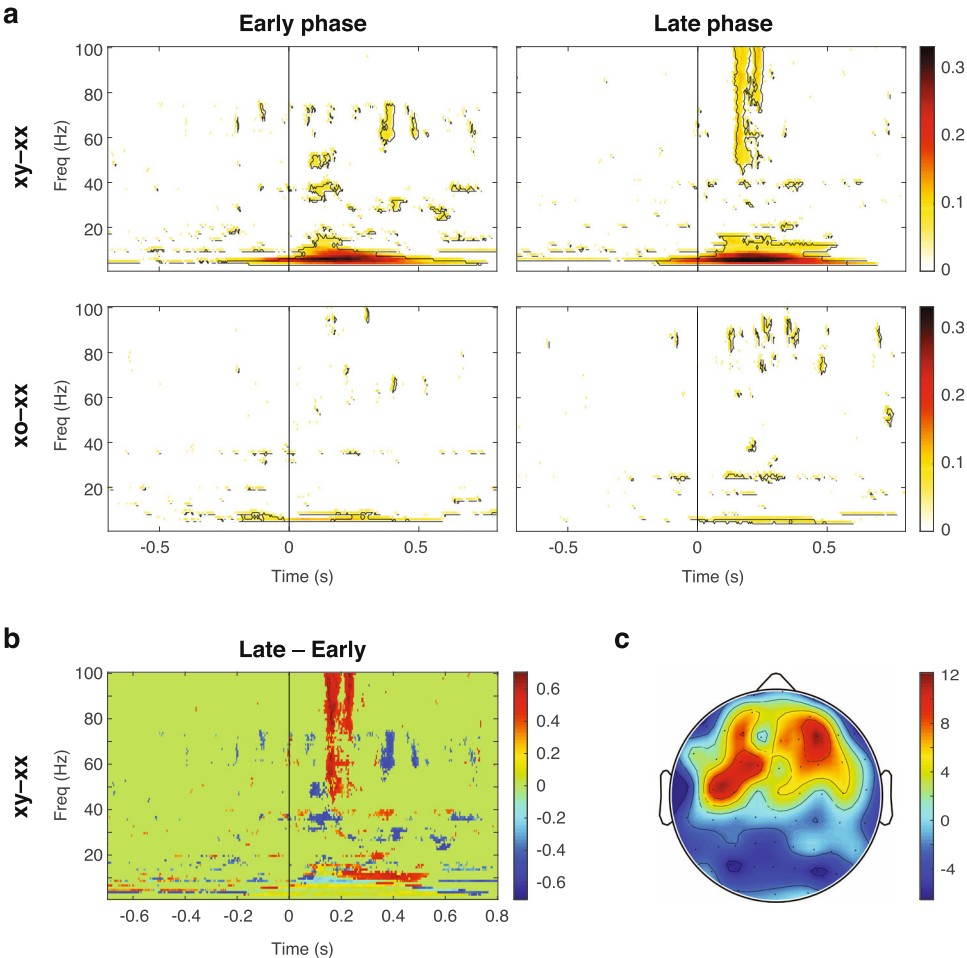

**Fig. 7 PE1 and PE2 during learning. a** The overall within-block contrast responses, xy – xx and xo – xx, in the early and late phases. The pixel value represents the ratio of significance across all channels and contrasts. **b** The first principal component to visualize the difference between the late and early phases (Late – Early) in the significant contrast responses (xy – xx). **c** The corresponding spatial location of the first principal component.

responses (xy – xx and xo – xx) in the early and late phases are shown in Fig. 7a. Stronger significant contrast responses were found in the late phase, particularly in the contrast xy – xx. This is consistent with the view that predictions became more precise in the later phase, thus the tone that violated local or global regularities induced a larger surprise and stronger PE1 and PE2 in the contrast responses.

To further visualize the learning effect in xy – xx, we compared the significant contrast responses (value = 0 or 1) between the early and late phases (Late – Early) for each channel, and performed PCA on the 2D data (*Channel × Time-Frequency*, value = −1, 0, or 1) (see "Methods"). The first and the most dominant principal component showed a stronger gamma-band response followed by a stronger alpha-band response during the late phase (Fig. 7b), which appeared primarily in the frontal and the frontocentral regions (Fig. 7c). Note that the across-block contrasts were excluded in this analysis, since learning varied across blocks due to different transition and sequence probabilities and thus cannot be distinctly examined.

**Interdependence of prediction and prediction-error signals.** The neural signatures of PE1, PE2, P1, and P2 were extracted based on the signal dependence proposed by the model. Based on the model, during the xx sequence in Blocks 1–4 (where xx is the dominant sequence), greater $P1_x$ will lead to smaller $PE1_x$ in the x stream, i.e., prediction reduces the surprise (Fig. 8a). However, greater $P1_Y$ will lead to greater $PE1_y$ in the y stream since $PE1_y$ is a

negative error (indicated as dashed arrows), i.e., strong prediction leads to a bigger surprise when the input is omitted. Therefore, $P1_x$ and $PE1_x$ are negatively correlated, while $P1_y$ and $PE1_y$ are positively correlated (indicated as the black negative and positive signs, respectively). Similarly, since $PE2_x$ and $PE2_y$ are negative errors, greater $P2_x$ and $P2_y$ will lead to greater $PE2_x$ and $PE2_y$, respectively (indicated as the red positive signs). Furthermore, smaller inputs $PE1_x$ and $PE1_y$ will lead to greater omission errors $PE2_x$ and $PE2_y$, respectively (indicated as the red negative signs). Collectively, the correlation between P1 ($P1_x + P1_y$) and PE1 ($PE1_x + PE1_y$) is not significant, since the correlations share different signs in the x and y streams. At the second level, P2 and PE2 are positively correlated in both streams, and PE1 and PE2 are negatively correlated in both streams. Collectively, the correlations between P2 ($P2_x + P2_y$) and PE2 ($PE2_x + PE2_y$) and between PE1 and PE2 are significantly positive and negative, respectively. On the other hand, the correlations are opposite during the xy sequence in Blocks 5–8 (where xy is the dominant sequence), since the negative and positive errors are switched.

To examine these theorized correlations in the EEG data, we monitored how PE1, PE2, P1, and P2 changed across trials, and examined whether and how their activations correlated with each other. The single-trial activations of PE1, PE2, P1, and P2 were obtained by projecting EEG responses from each trial onto their corresponding spatio-spectro-temporal structures (Fig. 8b). For PE1 and PE2, the spatial structures were created from the *Channel* dimension (Fig. 5a) and the spectro-temporal structures

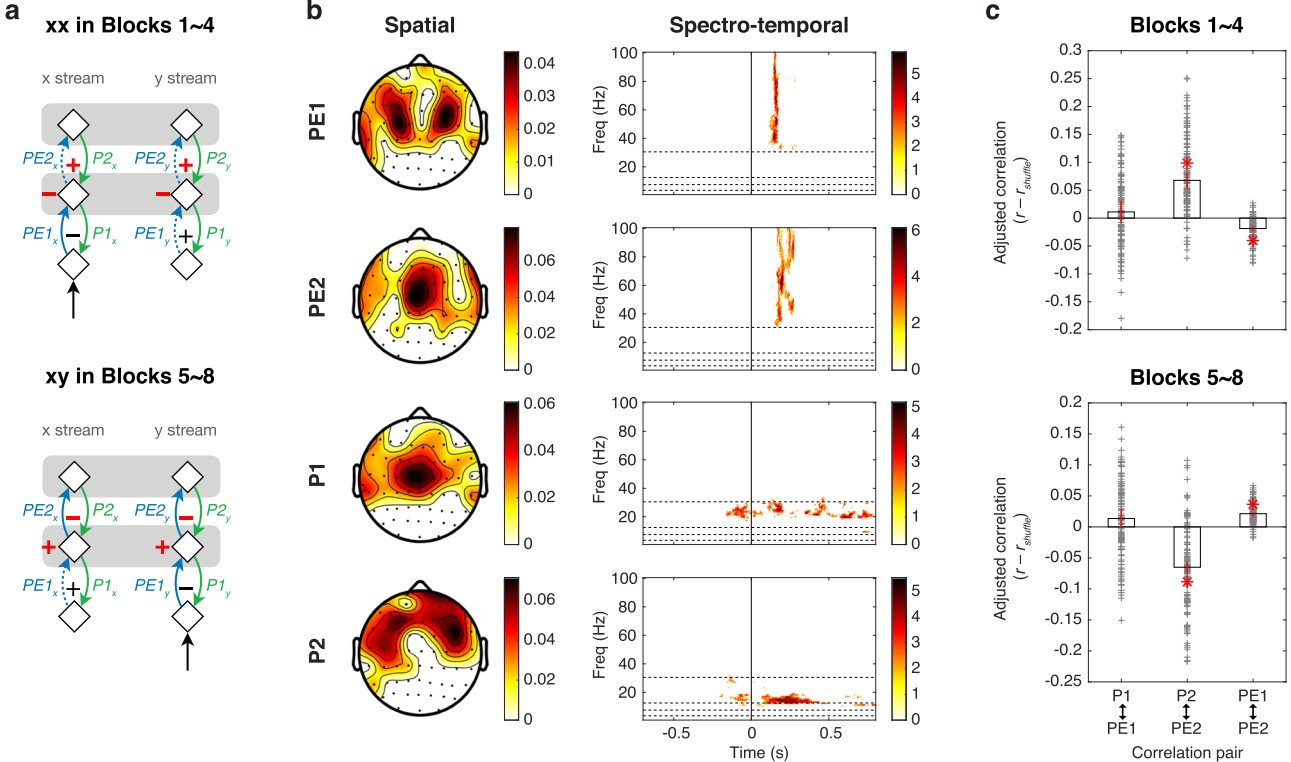

**Fig. 8 Correlated activations and theorized signal flow. a** The models of the xx sequence in Blocks 1–4 and the xy sequence in Blocks 5–8. Positive and negative signs indicate positive and negative correlations, respectively. Red positive and negative signs indicate same tendencies in both the x and y streams, and black signs indicate different tendencies across the x and y streams. **b** The spatial structures and spectro-temporal structures of PE1, PE2, P1, and P2. **c** The adjusted correlations ($r - r_{shuffle}$) in Blocks 1–4 and Blocks 5–8. For each pair, the median and the corresponding 95% confidence interval are shown as the black bar and red vertical line, respectively. All 120 data points (4 blocks * 30 subjects) are shown as gray crosses. Significant correlations are indicated by red asterisks (see details in the "Results" section).

were created from the gamma-band components in the *Time-Frequency* dimension (Fig. 5b). For P1 and P2, the spatial structures were created from *Channel* dimension (Fig. 6a) and the spectro-temporal structures were created from the beta-band components in the *Time-Frequency* dimension (Fig. 6b). To refine the spatial and spectro-temporal structures, the absolute values below the corresponding median values were set to zero. Note that P1 and PE2 components shared similar spatial distribution, which were consistent with the model in which the neural populations for P1 and PE2 locate at the same level (see Fig. 2).

The ERSP responses on each trial were then projected onto each spatio-spectro-temporal structure, which resulted in a scalar value for each structure (see "Methods"). These values indicate how much PE1, PE2, P1, and P2 appeared in the single-trial EEG responses (see an example in Supplementary Fig. 11). We further quantified the correlation coefficients ($r$) between the time courses of projection values to evaluate how PE1, PE2, P1, and P2 interacted with each other. To ensure the correlations were not artifactually caused by the spatio-spectro-temporal structures which shared overlapping features, we measured the projection values and the corresponding correlation coefficients for ERSP responses shuffled over channels, frequency bins, and time points (see Supplementary Fig. 11). The average correlation coefficients from 500 shuffles are denoted as $r_{shuffle}$. Figure 8b shows the adjusted correlations ($r - r_{shuffle}$) between three direct interactions: P1 ↔ PE1, P2 ↔ PE2, and PE1 ↔ PE2.

Among Blocks 1–4, significant positive and negative correlations were found between P2 and PE2 and between PE1 and PE2, respectively ($p < 1e−15$, Wilcoxon signed rank test, two-sided, $n = 120$: 4 blocks * 30 subjects) (Fig. 8c). Among Blocks 5–8,

significant negative and positive correlations were found between P2 and PE2 and between PE1 and PE2, respectively ($p < 1e−14$). These results were consistent with the model predictions.

## Discussion

We provide a quantitative definition of prediction and prediction-error signals based on the predictive coding theory, allowing us to extract hierarchical prediction and error signals from the neural responses. We also demonstrate that the hierarchical and bidirectional predictive coding framework was most suitable to describe the observed data. The utility of our computational strategy and its value for the field is that it can be applied to any experimental paradigm where predictability can be defined, not just the local-global paradigm.

In Fig. 9, we provide an analysis-driven model of the signal flow map for predictive processing in human EEG data for the local-global paradigm in a 2-level cortical hierarchy. The theoretical model illustrates the quantitative interactions between prediction signals at the lower level (P1) and the higher level (P2), and the prediction-errors signals at the lower level (PE1), and the higher level (PE2). The results provide a cohesive view of how these multiplicative signals propagate and interact in the cortex based on their timing and dependence. Here, we will describe below the signal information flow of a 2-tone sequence in the cortical hierarchical map after the transition and sequence probabilities are learned. We exclude adaptation for simplicity. The first tone initially evokes a sensory response (see step 1 in Fig. 9). Since the first tone is unpredictable due to the random interval between sequences, there is no P1 nor P2. Without

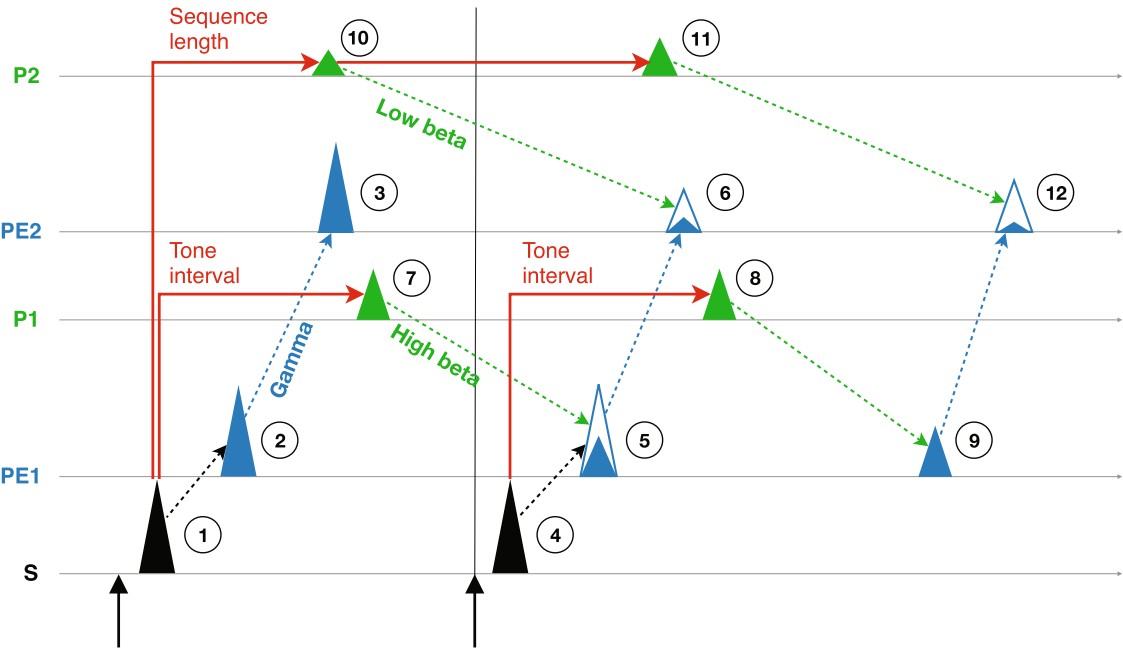

**Fig. 9 Theorized signal flows.** The theorized signal flows during a 2-tone sequence. See details in the "Discussion" section.

cancellation by these absent prediction signals, the sensory response generates PE1 after a delay (step 2), which further propagates and generates PE2 (step 3). These bottom-up signals are carried by gamma oscillations. Similarly, the sensory response to the second tone (step 4) generates PE1 (steps 5), which continues to propagate to generate PE2 (step 6). To confront PE1, we theorize that P1 predicts not only the size of incoming PE1 but also its timing (i.e., the tone interval), and is triggered by each tone (steps 1 and 4) and sent via top-down communication from Level 1 via high-beta oscillations (steps 7 and 8). Importantly, P1 needs to be activated before the sensory input so it can propagate from Level 1 to Level S to cancel it. Since the sensory response to the second tone is predicted by P1, the size of the consequent PE1 is reduced (step 5). For the last tone, there is no sensory input to confront P1 triggered (step 8), thus an omission error is generated (step 9), which represents the surprise of the sequence ending and is canceled at Level 2. To confront PE2, we assume that once the sequences are learned (not only the probabilities of xx, xy, and xo, but also the sequence length), the first tone is sufficient to trigger the prediction of the last tone (x, y, or o), its timing, and the subsequent ending. This assumption is supported by our previous study using the local-global paradigm in monkeys, where we found that global prediction signals occurred soon after the first stimulus[20]. Subsequently, P2 is transmitted from Level 2 via low-beta oscillations (steps 10 and 11) to reduce the sizes of PE2 from the last tone (step 6) and the omission at the sequence ending (step 12). We note that PE2 at step 12 could be very small since the sequence ending is highly predictable (although not 100% predictable due to the omission trial). The described signals represent their properties and flows in a single trial. In contrast, some signals will be canceled out in the contrast between trial types. For example, the error signals at steps 2 and 3 are the same in all sequences, thus are canceled out in the within-block contrasts, and PE1 and PE2 identified by the within-block contrasts (Fig. 5) correspond to PE1 at step 5 and PE2 at step 6, respectively. Furthermore, the error signals at steps 9 and 12 only occur in the xx sequence, since they are caused by the additional local prediction from the last x tone in the xx sequence. Therefore, in the within-block contrasts (xy – xx or xo – xx), and they could

underlie the negative gamma oscillations observed at longer latencies (see Fig. 5b, e). On the other hand, P1 and P2 identified by the across-block contrasts (Fig. 6) correspond to P1 at steps 7 and 8 and P2 at steps 10 and 11, respectively. Together, these results reveal the map of signal flow for predictive coding in the local-global paradigm. We will next discuss the properties, timing, and dependence of these signals within the network.

The prediction signals P1 and P2 appeared before the last tone (see results in Fig. 6e and our model in Fig. 9). This proactive emergence is an expected feature of prediction-related signals. In human, pre-stimulus alpha-band EEG oscillations were found to influence early stages of visual processing[50], and MEG signals encoded incoming predictable stimuli were observed shortly before they were presented[51]. In monkeys, enhanced beta-band functional connectivity was found before predictable stimuli[32]. Our results demonstrate that these proactive prediction signals occur across multiple hierarchies and set a baseline for incoming error information. How early prediction signals need to be activated depends on two factors. First, neuronal transmission delays that are required for signal propagation and neuronal processing, and second, the actual timing of the predicted events (e.g., the timing of the next tone and the end of the sequence). Neuronal transmission delays are often neglected in computational models of predictive coding[3,52,53], but they are critical for prediction and prediction-error signals to align properly across cortical hierarchies in real time[54]. However, empirical evidence of how the timings of predictions are tunned across hierarchies is lacking. On the other hand, event timing predictions have been widely studied by using temporal reproduction or foreperiod paradigms where subjects learn and anticipate the forthcoming sensory stimulus, and the neural substrates of temporal prediction have been characterized as the phase of alpha-band oscillations in EEG[50], the power of beta-band oscillations and the contingent negative variation in EEG[55,56], the phase-power coupling between theta and beta bands[57], and population neuronal firing rates in the dorsomedial frontal cortex[58]. However, it remains unclear how these proactive neuronal processes are initiated. In this study, we theorize that P1 and P2 were triggered by the sensory input, which suggests the existence of direct pathways that bypass

intermediate hierarchical levels. One possible route is the thalamocortical pathways, which converge with corticocortical pathways to enable higher auditory and visual perception[59,60].

Signals related to top-down predictions and bottom-up prediction errors are channeled by neural oscillations of distinct frequencies: alpha/beta and gamma bands, respectively[20,23,61–63]. This asymmetric signal transmission is anatomically plausible, since alpha/beta oscillations are largely found in the deep layers (5/6) of the cortex, whereas gamma activity is prominently generated in the superficial layers (2/3)[64–66]. This expectation is also functionally reasonable, since top-down signals could serve a modulatory integrative function operating over longer timescales, while bottom-up signals could require higher frequencies with greater energetic costs in order to achieve higher communication throughput[52,61,65]. Our results demonstrate that frequency specificity also occurs for prediction signals, where the frequency that channels prediction information is hierarchy level-specific. These hierarchy level-specific neural oscillations have also been found in bottom-up signals in monkeys, where slightly different gamma bands were found to carry feed-forward prediction-error signals at different hierarchical levels[20]. Functionally, the hierarchical ordering of neural oscillations, or frequency ordering, could allow different levels in hierarchies to encode information at different time scales, where lower and higher levels can predict faster and slower dynamical events, respectively. Anatomically, different brain areas naturally resonate at particular frequency bands[67,68], which could be the neural basis of such frequency-specific predictions. Particularly, the intrinsic timescales of neuronal activity have been found to be hierarchically organized, with sensory and prefrontal areas showing shorter and longer timescales, respectively[69,70]. This organization may allow lower levels to quickly and robustly track dynamic sensory inputs, while higher levels can integrate multi-scale information and achieve noise-invariant computation.

Prediction errors can be either positive or negative, since bottom-up inputs can be larger than predicted (e.g., when receiving an unexpected stimulus) or smaller than predicted (e.g., when failing to receive an expected stimulus). It has been proposed that positive and negative errors are processed by distinct neurons in the neocortex, since cortical baseline firing rates are low and it is less plausible for a single neuron to signal both types of prediction errors by changing its firing rate bidirectionally[71]. However, most studies only focused on positive or negative prediction errors[5,72–75], possibly due to the difficulty of separating neuronal processes underlying positive and negative error computations which can occur simultaneously. Using a model-fitting approach, we showed that EEG responses capture both positive and negative prediction errors. To identify the positive and negative error signals or test the theorized biological circuitries[71,76], one would need to use neural recordings with single-cell resolution, such as single-unit activity recordings or calcium imaging. Another solution would be to design a task in which positive and negative prediction errors are generated in different and distant domains, such as probing expectations of face and place stimuli in different brain regions[19].

We observed reduced responses during sequences with repetitive stimuli (see Supplementary Fig. 6), thus we hypothesize that neural adaptation occurs for those sequences and P1 predicts a reduced input scaled by an adaptation factor. Our model does not differentiate the origins of this presumed adaption, only its outcome. One possible cause is stimulus-specific adaptation (SSA), which is a lower-level inhibitory neuronal mechanism in response to repetitive stimulation that has been observed in both cortical and subcortical structures[77–79]. Another possible cause for the adaptation is predictive coding itself, where the prediction of transitions between identical tones is learned during repetitions,

and the repetitive tones generate less surprise over time. To fully explain the data will require a model that includes the interplay between predictive coding and SSA to describe the neural dynamics during each tone in cortical and subcortical areas.

Our model focuses on signal dependence after transition and sequence probabilities are learned and errors are minimized. To understand the dynamic process of prediction updating and error minimization, it is essential to examine how probabilities are encoded. It is thought that probability distributions, or their log values, are encoded straightforwardly in population firing rates (as adopted in our model), combinational firing patterns of neuronal populations representing specific probability distributions (called basis functions), or the value of membrane potentials[40], and their updates based on prediction errors are mediated by neuromodulators, such as acetylcholine[80]. One candidate to incorporate these ideas is a Bayesian model called the Hierarchical Gaussian Filter[81], which updates predictions by precision-weighted prediction errors[1,3,12] and was implemented to examine prediction-error signals during learning in the brain[74,82–85]. However, its implementation in hierarchical prediction is limited (despite the term hierarchical in the name, which refers to a motor part of the model) but highly demanded. One important feature in our model is that while prediction is established in each individual stream, its value is determined by the stimulus probability (TP at Level 1 and SP at Level 2) which requires information integrated from both streams. Therefore, we believe that prediction is encoded in a neuronal network with four key features which can be tested by using finer-grained measurements such as single-unit activity recordings or calcium imaging: (1) inter-stream connections (spans spatially across streams), (2) probability encoding (changes activation based on sensory predictability), (3) proactive timing (activates before the sensory input), and (4) top-down regulation (influences responses at the lower hierarchy). Our EEG results showed an enhanced frontal alpha-band response during the late phase of learning (Fig. 7b), which could represent a prediction update process that occurred immediately after gamma-band prediction-error signals. Similar long-latency alpha-band activities in the frontal cortex have been observed in ECoG in both humans and monkeys[20,22]. Furthermore, alpha-band signal magnitudes have been found to correlate with prediction updates when changes in the stimulus probability occur[35,86]. Understanding interactions among alpha-band activity, gamma-band prediction-error, and beta-band prediction signals, may require the use of a trial-by-trial analysis with Bayesian modeling[87,88].

In summary, we used a cortical signal dependence model to disentangle prediction and prediction-error signals and reveal a frequency ordering of prediction signals that allows different hierarchical levels to encode information at different time scales in the human brain. These results advance the physiological measurement and modeling of predictive coding, and provide a platform to examine predictive signaling beyond two hierarchical levels (e.g., information of longer timescales or greater abstraction) and among multiple sensory modalities in normal and disordered brain.

## Methods

**Model calculation**. We propose a simple model where the optimal value of each prediction signal is to minimize the mean-squared error received. For example, the mean squares of $PE1_x$ (denoted by $MSPE1_x$) can be devised as (based on the bar graph in Fig. 2a):

$$MSPE1_x = TP_x * (s_0^{n-1} - P1_x)^2 + (1 - TP_x) * (P1_x)^2 \qquad (1)$$

The minimums occur when:

$$\frac{d}{dP1_x} MSPE1_x = 0 \tag{2}$$

Which leads to:

$$P1_x = s_0^{n-1} * TP_x \tag{3}$$

And $P1_y$ can be obtained in the same fashion:

$$P1_y = TP_y \tag{4}$$

This represents the optimal prediction where first-level prediction errors are minimized. Then we added the scaling factor $s_1$ to $P1_x$ and $P1_y$ and calculate the mean squares of $PE2_x$ (denoted by $MSPE2_x$):

$$MSPE2_x = SP_{xx} * (|s_0^{n-1} - P1_x * s_1| - P2_x)^2 + (1 - SP_{xx}) * (P1_x * s_1 - P2_x)^2 \tag{5}$$

The minimums occur when:

$$\frac{d}{dP2_x} MSPE2_x = 0 \tag{6}$$

Which leads to:

$$P2_x = SP_{xx} * (|s_0^{n-1} - P1_x * s_1|) + (1 - SP_{xx}) * P1_x * s_1 \tag{7}$$

And $P2_y$ can be obtained in the same fashion:

$$P2_y = SP_{xy} * \left(|1 - P1_y * s_1|\right) + \left(1 - SP_{xy}\right) * P1_y * s_1 \tag{8}$$

Note that the $P2_x$ and $P2_y$ here represent the optimal predictions when potential erroneous predictions at the first level are considered. Also, $s_2$ was applied to calculate the second level prediction errors, i.e., $P2_x*s_2$ and $P2_y*s_2$ were used (as shown in Fig. 2d).

Based on the model, all prediction signals are determined once the transition probabilities ($TP_x$ and $TP_y$), sequence probabilities ($SP_{xx}$ and $SP_{xy}$), and scaling factors ($s_0$, $s_1$, and $s_2$) are known. The transition probabilities can be calculated based on the number of tones in a sequence ($n$) and the sequence probabilities. For $n$-tone sequences, there are $n-1$, $n-2$, and $n-2$ transitions from tone $x$ to $x$ in the xx, xy, and xo sequences, respectively. Combining with the corresponding sequence probabilities, the expected number of transitions from tone $x$ to $x$ (denoted by $TN_x$) is

$$TN_x = (n - 1) * SP_{xx} + (n - 2) * SP_{xy} + (n - 2) * (1 - SP_{xx} - SP_{xy}) \tag{9}$$

Similarly, the expected number of transitions from tone $x$ to $y$ (denoted by $TN_y$) is:

$$TN_y = 1 * SP_{xy} \tag{10}$$

For the expected number of transitions $x$ to $o$ (denoted by $TN_o$), the transition from tone $x$ to $o$ at the end of the xx sequence is considered:

$$TN_o = 1 * SP_{xx} + 1 * \left(1 - SP_{xx} - SP_{xy}\right) = 1 - SP_{xy} \tag{11}$$

Thus, $TP_x$, $TP_y$, and $TP_o$ can be calculated as

$$TP_x = \frac{TN_x}{TN_x + TN_y + TN_o} = \frac{TN_x}{TN_x + 1} \tag{12}$$

$$TP_y = \frac{TN_y}{TN_x + TN_y + TN_o} = \frac{TN_y}{TN_x + 1} \tag{13}$$

$$TP_o = \frac{TN_o}{TN_x + TN_y + TN_o} = \frac{TN_o}{TN_x + 1} \tag{14}$$

The values of these transition probabilities in the 8 blocks are shown in Fig. 1b. Examples of the strengths of the prediction and prediction-error signals in Blocks 3 and 7 with $s_0 = s_1 = s_2 = 1$ (the optimal predictions with no sensory adaptation) are shown in Fig. 2e. The MATLAB code for these calculations is also provided.

**Participants**. Thirty healthy adults were recruited in this study (15 males and 15 females; age: $24 \pm 2.6$ years old, mean ± standard deviation). The inclusion criteria for participants were: (1) aged 20–40 years old; (2) no participation in drug studies; (3) no apparent cognitive difficulties or serious deficits in vision and hearing; (4) no known neurological and psychological diagnosis. All research protocols were approved by the Research Ethics Committee of the National Taiwan University Hospital (201906081RINA). Each participant signed informed consent before the experiment.

**Stimuli**. Two tones were created by combining three sinusoidal waves of different base frequencies: 350, 700, and 1400 Hz for the low-pitch tone (tone $A$), and 500, 1000, and 1500 Hz for the high-pitch tone (tone $B$). The duration of each tone was set to be 100 ms with a 7 ms rise and fall. A tone sequence was composed of either 2 or 3 tones in which 200 ms was set between successive tone onsets within a sequence, and 1000–1400 ms was set between the offset of the last tone of a sequence and the onset of the first tone of the following sequence (see Fig. 1a).

Eight sequence blocks were used, each with a total of 144 sequences (see Fig. 1b). The order of the sequences was pseudorandom within each block, where the total sequences were divided into four phases while each phase kept the same sequence ratios. For example, for Block 1, each phase had 24 trials of xx, 6 trials of xy, and 6 trials of xo (a total of 36 trials in a phase). The sequence order was randomized for each phase in each block, with possibilities of consecutive rare sequences (e.g., two consecutive xy sequences in Block 1). The reason for the pseudorandom order was to maintain overall sequence probabilities throughout the learning. Furthermore, the reason to allow consecutive rare sequences was to avoid introducing additional statistical structures into the sequences. Each block was delivered twice, one time with tone $A$ as the frequent tone (block $A$) and the other time with tone $B$ as the frequent tone (block $B$). For example, for Block 1 (see Fig. 1b), sequences $AA$, $AB$, and $AO$ ($O$ as omission) were used in one run, and sequences $BB$, $BA$, and $BO$ were in the other run. A total of 16 runs of 144 sequences were used.

**Experimental procedure**. Each participant underwent 16 blocks in a pseudorandom order. During a block, participants were instructed to visually fixate at a central fixation cross on the screen and pay attention to the sounds. To minimize the chance that the learned sequential structure of the previous block being carried over to influence the next block, participants were presented with a 15 s video during breaks between successive blocks for wash-out purposes. All experimental protocols were programmed with the MATLAB-based Psychophysics Toolbox Version 3[89] and all auditory stimuli were delivered through a pair of desktop speakers (~60 dB).

**EEG recording**. EEG signals were recorded with an elastic custom EEG cap (64-channel Quick-cap, the extended 10–20 system, Compumedics Neuroscan, Australia) and a SynAmps RT amplifier (Compumedics Neuroscan, Australia). EEGs were on-line referenced to the reference electrode near Cz. At the preparation stage before recordings, electrode impedances were kept to be <2 kΩ for the left and right mastoid (M1 and M2) electrodes, 10 kΩ for the eye electrodes, and 5 kΩ for the remaining electrodes.

During the EEG recording, eye blinks and eye movements were detected by horizontal and vertical electrooculography (HEOG and VEOG) electrodes. The HEOG electrodes were attached to the outer canthi of each eye to monitor horizontal eye movement. The VEOG electrodes were attached to the supraorbital and infraorbital ridge of the left eye to monitor vertical eye movement and eye blinks. When each tone sequence was displayed, the onset time of the first tone was simultaneously labeled as an event code, a number representing types of tone sequences in different blocks. Raw EEG and EOG signals were recorded online with a band-pass filter of 0.01 to 100 Hz, a gain setting of 1000 and digitalization, and by a sampling rate of 500 Hz. Then all EEG data were digitally stored for later off-line preprocessing. The whole recording process took place in a sound-attenuated, dimly lit room.

**EEG analysis**

*EEG preprocessing*. EEG preprocessing was done by EEGLAB on MATLAB[90]. The raw data were first re-referenced to the average of the left and right mastoids (M1 and M2) to eliminate systematic noise from the environment. Then EEG epochs were extracted from –1.5 to 2.3 s for the 2-tone sequences and from –1.5 to 2.5 s for the 3-tone sequences (time zero as the onset of the first tone in a given sequence). This segmentation keep data from –1.2 s before the first tone to 1.9 s after the last tone for both 2-tone and 3-tone sequences. Excessive fluctuations or high-frequency noise in the EEG epochs were eyeball screened and manually rejected. For each participant, an average of ~2.6% of the total 2304 trials (144 trials per block, 8 blocks, 2 configurations: block A and block B) were rejected ($60.9 \pm 73.9$ trials, mean ± standard deviation).

To remove eye movement artifacts from the signals, we first performed an independent component analysis (ICA) with the infomax algorithm (pop_runica.m) with the electrodes VEO and HEO removed (62 channels left), and used the ADJUST algorithm to automatically identify and remove artefactual component(s) related to eye movement[91] (pop_ADJUST_interface.m). To acquire reference-free signals, a 3D 60-channel EEG montage spherical coordinates were used to estimate scalp current source density (CSD), where the cerebellar electrodes CB1 and CB2 were excluded. The CSD analysis was done by using the CSD toolbox with a smoothing constant lambda of 1e−5 and the head radius of 10 cm[44].

*Event-related spectral perturbation (ERSP)*. ERSP was calculated for each trial type (e.g., the xy sequence in Block 3) and each subject. For each subject, channel, and trial, the time–frequency representation (TFR) of the CSD signal was generated by Morlet wavelet transformation at 100 different center frequencies (1–100 Hz) with the half-length of the Morlet analyzing wavelet set at the coarsest scale of 7 samples, which is implemented in the FieldTrip Toolbox (ft_freqanalysis.m)[92]. Baseline normalization was then performed to calculate the decibel values by using the baseline period from –0.2 to 0 s (time zero as the onset of the first tone) (ft_freqbaseline.m). For each trial type, the ERSP was calculated by averaging the normalized TFRs from the corresponding trials including both block A and block B to eliminate tone-specific effects.

*Contrast response.* For each contrast (use xy – xx in Block 3 as an example), the contrast response was calculated for each channel across 30 subjects. The average ERSPs for the xy and xx sequences were first calculated across 30 subjects, and then the difference in the average ERSPs was obtained. To measure the significance of the difference (as the black contours shown in Fig. 4a, b), the confidence intervals of each ERSP value for the xy and xx sequences were first obtained by bootstrapping the corresponding ERSPs from 30 subjects 1000 times ($\alpha = 0.05$, two-sided, false discovery rate correction). By comparing the confidence intervals, the significant contrast responses were obtained (value = 0 or 1, where 1 represented significance). For the decomposition analysis, the contrast responses were masked with the significance, where nonsignificant values were set to 0.

*Overall occurrence of significant contrast responses.* The overall occurrence of significant contrast responses shown in Supplementary Fig. 1 was obtained by averaging the significant contrast responses (value = 0 or 1) from multiple contrasts and all channels. For each frequency bin, the standard deviation of the average significance during a baseline period (–0.5 s ~ –0.2 s for 2-tone sequences, –0.7 s ~ –0.4 s for 3-tone sequences, where time 0 represented the onset of the last tone) was calculated, and only values above 5 times of the standard deviation were shown for clarity.

*Data-driven analysis with parallel factor analysis (PARAFAC).* To obtain a comprehensive view of the contrast responses, the contrast responses masked with the significance were pooled to create a tensor with three dimensions: *Channel* (brain area), *Time-Frequency* (in-trial dynamics), and *Contrast* (contrast response), for the anatomical, dynamic, and functional aspects of the data, respectively. The dimensionality of the tensor was 60 (channels) by 37,500 (375 time points and 100 frequency bins) by 16 (within-block contrasts). To extract structured information from the dataset, we factorized the 3D tensor into multiple components by performing PARAFAC, a generalization of principal component analysis (PCA) to higher-order arrays[41], which was previous used for the computational extraction of latent structures in functional network dynamics[20,38,39]. PARAFAC was performed using the N-way toolbox[93], with no constraint on all three dimensions (parafac.m). The convergence criterion (i.e., the relative change in fit for which the algorithm stops) was set to 1e−6. The initialization method was set to be direct trilinear decomposition (DTLD), which was considered the most accurate method[94]. To determine the number of structures hidden in the dataset, we performed the core consistency diagnostic (CORCONDIA) to identify the appropriate latent structures where adding other latent structures does not considerably improve the model fit[42].

*Model-fitting with PARAFAC.* To decompose pooled contrast responses into components with theorized contrast values, PARAFAC was performed with the third dimension *Contrast* fixed with the values proposed by the model (using FixMode and OldLoad inputs in parafac.m). For any given model (e.g., within- or across-block contrasts, different adaption factors, different error types, etc.), a core consistency diagnostic value and residual sum of squares (RSS) were obtained to represent how well the pooled data fit the model.

*Model comparison.* The Bayesian information criterion (BIC) was calculated to evaluate the goodness of fit of each model. For tensor-based decomposition analysis, such as PARAFAC, BIC was calculated as follow[95]:

$$BIC = u * \log\left(\frac{RSS}{u}\right) + w * \log(u) \tag{15}$$

where $u$ represents the number of data point, $w$ represents the number of estimated elements, and RSS represents the residual sum of squares from PARAFAC. The values of $u$, $w$, and RSS for each model are shown in Table 1.

*Visualization of learning in contrast responses.* To visualize the learning effect in xy – xx (as shown in Fig. 7b, c), the differences in significant contrast responses between the early and late phases (Late − Early) were obtained for each channel (value = −1, 0, or 1). A two-dimensional data was then created: 60 (channels) by 37,500 (375 time points and 100 frequency bins), and PCA was performed on the 2D data for visualization.

*Single-trial projection and adjusted correlation.* The projection value of a single-trial ERSP responses (ERSP = 60 channel × 100 frequency bins × 375 time points) on to the spatial structure (S = 1 by 60) and the spectro-temporal structure (F/T = 100 by 375) of a component (shown in Fig. 8a) was calculated as S*ERSP*F/T, which yields a single scalar value. For all the available trials (after removal of bad trials) in each block (a total of 480 blocks: 8 block types, 2 runs per block types, and 30 subjects), the Pearson correlation coefficients ($r$) were calculated between the 144-value time courses of PE1, PE2, P1, and P2. To shuffle ERSP response, values in channel, time, and frequency are randomly exchanged. For each shuffle, the projection values and the correlation coefficients were calculated as described above. The average correlation coefficients across 500 shuffles ($r_{shuffle}$) are measured, and the adjected correlations were calculated as $r − r_{shuffle}$. See Supplementary Fig. 11 for an example of the process.

*Statistics and reproducibility.* The sample size is comparable to previous similar EEG/MEG studies[25,27], and no participant was excluded. The proposed model is fully described in equations and the MATALB code for its calculation is provided. For EEG analysis and data-fitting analysis, the details including the variable dimensionality, MATALB toolboxes, functions, and key parameters are provided. For statistical comparisons, details including the number of resampling and multiple comparisons methods are provided. The only subjective step is the EEG preprocessing where bad trials were manually excluded via visual inspection. However, we followed a general guideline and only ~2.6% of the total 2304 trials were excluded.

**Reporting summary**. Further information on research design is available in the Nature Research Reporting Summary linked to this article.

## Data availability
Source data underlying main figures are presented in Supplementary Data 1. The raw EEG data are available from the corresponding author upon request.

## Code availability
The code to calculate values of predictions and prediction errors in the proposed model has been deposited in Zenodo[96].

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

## Acknowledgements
We thank Rufin VanRullen for the critical comments, and Charles Yokoyama for valuable discussion and paper editing. We also thank Hsuan-Chi Liu, Chin-Kun Fu, and Shih-Yao Mao for helping with participant recruitment and experiment preparation. This work was supported by World Premier International Research Center Initiative (WPI), MEXT, Japan (to Z.C.C.), and the Ministry of Science and Technology of Taiwan (MOST 106-2420-H-002 -008-MY2 and MOST 109-2410-H-002-106-MY3) (to C.W.).

## Author contributions
Z.C.C. conceptualized the study. Y.T.H., C.W., and Z.C.C. refined the experimental protocol, and Y.T.H. collected the data. C.W. supervised the data collection. Z.C.C. proposed the theoretical model and performed the data analysis, and Y.T.H. performed data preprocessing. Z.C.C. wrote the first draft of the paper, and Y.T.H. and C.W. helped with editing. All authors contributed to and have approved the final paper.

## Competing interests
The authors declare no competing interests.
