## [Peer Review File · Communications Biology]

Reviewers' comments:

Reviewer #1 (Remarks to the Author):

This manuscript presents a computational framework to disentangle prediction and prediction error signals from EEG measurements during a hierarchical local-global auditory task. The authors designed a 3-level predictive coding model that can learn and recognize temporal regularities and sequences of tones presented in the experimental task. The sensory level is where the model received the input, whereas levels 1 and 2 are responsible for the recognition of the local and global patterns of auditory sequences in each experimental block. Dynamics of prediction and prediction error signals at each hierarchical model level are calculated based on the predefined (for each block) sequence structure (number of tones) and occurrence (distribution of xx, xY and xo sequences). This information is used to extract respective neural signatures corresponding to the prediction and prediction errors.

With this setup, the authors show that the brain use different frequencies to propagate information up and down in the hierarchy. Specifically, the gamma-band is associated with the bottom-up prediction errors, whereas the beta-band relates to the top-down predictions. These results nicely complement a number of studies that suggest frequency specialization for the bottom-up (gamma) and top-down (beta) information flows. The results further suggest that predictions about local and global regularities in sensory stimuli are propagated with different beta-bands: higher and lower frequency ranges respectively. That is the notion of "frequency ordering" in hierarchically organized stimuli, "requires" nested predictions.

The study provides an interesting approach that can be useful for identifying interactions between multiple processing hierarchies. If the framework could be extended to other modalities or scaled up for more complex sensory signals, it could become an important tool to understand hierarchical processing in the brain. Although the fitting results are overall consistent with existing neurophysiological evidence for predictive coding, the manuscript could be improved by providing additional information to enhance the biophysical plausibility of the model. In particular:

- The model relies on specific hypotheses about the form of prediction and prediction error signals, i.e. the prediction error neuron performs a subtractive operation, whereas the prediction neuron receives no direct input but outputs the same prediction signal once in the steady state. While this might apply for the particular paradigm used in the study, I would hesitate to generalize it to another setting, where e.g. one could also consider spike patterns, etc. Could the authors refer to existing literature to provide rationale of these very specific hypotheses?

- What is the neurophysiological basis of the construction of separate X and Y streams, and for the form of input (1 for the target, 0 for anything else)?

Since the model fitting indirectly relies on the contrast between different conditions, fundamentally different model configurations, such as without predictive coding or predictive routing (Bastos et al. 2020, PNAS), could presumably reach similar fitting results. Therefore, it would be important to either provide solid physiological evidence for the basic hypotheses or to compare between radically different hypotheses (not just slight variation of the current model).

Some detailed comment/questions:

- Line 7-10, page 3: I would suggest to also indicate that the predictions are updated to minimize prediction errors. Otherwise, it reads as prediction and prediction errors are "just" generated without any goal.

- Line 12-19, page 4: the authors mentioned a few examples of the correlation between (changes in) beta/alpha band activity and (changes in) prediction signals. How could the result of the current model provide mode specific explanations to such correlations?

- L8-22, P6: it is not clear that each block contains either 2-tone or 3-tone sequences (and not mixed 2 and 3 tone sequences). Although one could conclude it from e.g. diagram in Fig 1, mentioning it in the main text would avoid confusion about whether xxo (3-tone with emission) and xx (2-tone) sequences can occur within the same block and whether they are differently treated during analysis.

Furthermore, in the Methods section, it is mentioned that each block was delivered twice, one time with tone A and the other time with tone B. However, in the main text it is not clear whether the

authors treat predictions for tone A and tone B differently (e.g. P1x for tone A, is the same as P1x for tone B? (same goes for P2x, P1Y, P2Y)).

- L13, P9: it would be useful to provide discussion about how the minimization of mean-square error might be biophysically implemented.

-L15, P10: although the authors showed later that the model fitting is better when an additional x-o transition is considered at the end of an xx sequence, such construction changes the expected length of the particular sequence hence the temporal prediction. If this is considered in the contrast response, the model fit may be affected.

-L3, P12 and Fig 3B: why are there still negative values in the POS condition and positive values in the NEG condition? It is unclear to me what is plotted in these figures, z-scores? Also, for all three figure panels, PE1 and PE2 are very similar. How should we interpret such results?

-L15, P14: it seems that the text should have pointed to Figure 3B (and not Figure 2B).

-L7-8, P15, Fig 4E-F: what are the functional interpretations of these areas? PE1 shows a slightly higher activation in the left compared to the right temporal area. Does this indicate auditory processing?

-L20-21, P16: as the authors pointed out here, PE1 and PE2 are always highly correlated even with a lower adaptation factor (is it true that the similarity between PE1 and PE2 is correlated with the adaptation factor?). This confound does not seem to be considered in the analysis of within-block contrast that separated PE1 and PE2.

-L23, P16: why only consider equal contribution of PE1 and PE2? Fitting $PE1 + p*PE2$ might provide different results.

L20-25, P17: Authors indicate that predictions of faster and slower events are encoded in fast frequency (~ 23 Hz beta) and slow (~ 15 Hz beta) bands. These results are derived from averaged neural signatures of predictions from 2-tone and 3-tone sequences. It would be interesting to see what frequency bands would appear for fast and slow events if one only considers either 2-tone or 3-tone sequences. Would the frequency ratio of slow/fast predictions follow the temporal regularity of slow-fast events in the stimuli (e.g., the difference between frequency bands to encode fast/slow events would be higher for 3-tone sequences)?

-L20-22, P18: As subjects learn x, they also learn about the violations, which may result in smaller surprise for xY in steady states. It might be useful to fit different models to the learning vs. the steady phase and compare the parameters.

-L21-22, P22: temporally unpredictable, but the tone identity is still predictable, therefore it might not be the case of "no prediction".

Additional general questions:

-Why was an adaptation factor applied to the prediction error signal, but not to the prediction signal?

-How would information encoded in Level 2 be read out from a presumably higher-level neuron to infer the input sequence?

In addition, a rather "cosmetic" suggestion would be to re-evaluate notations and naming of sequences. The authors did a formidable job including all sequence information in the names (e.g. xY22), but a bit more consistency (why Y is with a capital letter and x not: then, it gets confusing later on, with capital letters of PE and P, TP, etc in math equations) and use of superscripts and subscripts (e.g. superscript to indicate the model level and subscript to indicate sequence type, thus $P(1)(x)$ instead of P1x) would increase the readability of the text.

Reviewer #2 (Remarks to the Author):

Chao et al. have produced a remarkable advance, using a quantitative model to predict EEG responses responsible for prediction signals. The authors provide a quantitative definition of prediction error and prediction signals in the local/global paradigm with different manipulations of stimulus probabilities. This allows them to identify the spatio-spectral-temporal signatures of prediction. The work is excellent, technically sound, and provides a substantial advance. I have only one major concern and a

few minor textual suggestions.

Major Concern

Is the adaptation model by itself better or worse at explaining the EEG data than the PE1/PE2/P1/P2 model? In other words, could the authors quantify the amount of explained variance in their spatio-temporal-spectral data that is accounted for by adaptation alone, compared to the predictive coding model (and perhaps the predictive coding model considering only PE1 and P1)? The best way to make this comparison would be to use Bayesian model comparison, where the larger number of model parameters of the first and second level predictive coding models would penalize the model fit (forcing these models to explain more variance if they are to win as the "best" model). If the authors would be able to show this, it would show that a model consisting of top-down and bottom-up factors explains the data better than adaptation alone.

Minor Concerns

The authors state in the introduction that previous studies have not mapped prediction signals, although as later acknowledged in the discussion, there is previous work. In particular, human fMRI (Summerfield et al., 2006, reference 7) have shown frontal to posterior connectivity consistent with prediction signals, and pre-stimulus activity was found in monkeys by Bastos et al., 2020 (reference 32 in the manuscript). One missing evidence for neural encoding of prediction in spiking activity is the work of Bell, Ungerleider, et al., *Current Biology*, 2016. However, it is true that the literature has focused mostly on prediction error signals and the prediction signals themselves have been understudied. Please revise the introductory text to acknowledge this. It does not take anything away from the present work, which is still quite an advance, because it shows quantitatively how different types of prediction are encoded in the spatio-temporal-spectral domain.

Page 24, line 24, "velocity information" is used in an awkward/non-standard way, difficult to parse. Please re-phrase.

Page 25, lines 16-18, in referencing gamma/beta to superficial/deep layers, the authors do not cite any empirical papers. The appropriate reference here is Buffalo, Fries, Desimone et al., 2011 PNAS.

Page 25, lines 18-21, this distinction in terms of prediction errors using faster frequencies than the predictions that generate them was also a theoretical feature of the predictive coding model put forth by Bastos et al. (reference 42).

Could the authors make the code for determining the PE1/PE2/P1/P2 signals in the local/global oddball paradigm publicly available for download? That would help other authors to be able to also use this quantitative framework and would greatly contribute to advancing the field.

Reviewer #3 (Remarks to the Author):

Chao et al. measure brain responses while presenting auditory stimulus sequences, with the aim of finding neural correlates of predictive coding. They are specifically targeting correlates indicating prediction (as opposed to correlates of prediction error transmission, which have been reported in numerous studies). By a combination of tensor decomposition analysis and neural modeling, they report having found hierarchical prediction signals in the human brain. These prediction signals are reported to be "frequency-ordered" in the sense that the high beta frequency band represents tone-to-tone transition, and low beta band represents multi-tone sequence structure (and then gamma band reflects prediction error).

Researchers have tried to find correlates of prediction and dissociate these from prediction error signals for many years. The key towards achieving this is to use a paradigm that allows for a clear distinction between prediction and prediction error. Unfortunately, the paradigm used in the current study does not fulfill this requirement in a more convincing manner than previous studies have done. In fact, the authors do not explain the underlying logic of their paradigm. Instead they create the impression that they use a well-validated "local-global paradigm", with reference to Bekinschtein et al. (2009) and to their own prior work in primates (Chao et al., 2018, Neuron). The scientific strength of this local-global paradigm is that local predictions in 5-tone sequences are played against global predictions in terms of expecting A or B as the final tone of a sequence starting with AAAA.. In the current study, no 5-tone sequences are employed but only 2-tone sequences. Hence by all common knowledge on how the brain extracts auditory patterns, there will be no local prediction in the second position. The allusion to the local-global paradigm is thus misleading. This would be only an issue of framing if the authors' paradigm was convincing in and of itself, but it is really not obvious how their paradigm solves the dissociation of prediction and prediction error. On the contrary, there are some problematic assumptions in the paradigm, such as that there can be only neuronal adaptation for tones that appear twice in a sequence (ignoring the fact that adaptation can take place across sequences, i.e. spanning longer timescales). The authors also implicitly assume that the brain only ever makes one prediction, whereas some studies suggest that two opposing predictions can be maintained in parallel. Furthermore, essential details about sequence construction are missing, such as whether rare events (eliciting prediction error) can immediately repeat or not. (If they can't, this changes predictability in the stimulus right after the prediction error.) It remains unclear which contrasts really indicate prediction. One implicit assumption is that predictions must be learned across the block (nothing can be predicted at block start). This assumption would naturally lead to a modeling framework that specifically looks for a neural process that emerges during the course of the block (with sequence exposition). In contrast, the modeling approach identifies components that are present throughout the whole block, and then analyzes whether they change in amplitude between early and late portions (which gives a rather small effect, no striking evidence of emerging prediction). Hence altogether, although the study uses technically impressive approaches towards modeling, the underlying paradigm seems too weak to yield conclusive results.

There are further, more minor details, such as inconsistencies in the description: "During the task, participants passively listened to a series of short tone sequences based on the local-global auditory paradigm while brain activity was recorded by 64-channel EEG. To ensure vigilance, participants were instructed to both visually fixate and attend to the sounds." Participants cannot "passively listen" and "attend" to the same sounds at once. This is a critical difference as it informs about the top-down contribution in acquiring predictions. Since the paradigm is already tipped towards more global type of predictions (i.e., local stimulus-driven prediction contributions are weak, see above), heavy attention involvement would tip this balance even further and question the generalizability towards other predictive coding instantiations.

Reviewers' comments:

Reviewer #1 (Remarks to the Author):

This manuscript presents a computational framework to disentangle prediction and prediction error signals from EEG measurements during a hierarchical local-global auditory task. The authors designed a 3-level predictive coding model that can learn and recognize temporal regularities and sequences of tones presented in the experimental task. The sensory level is where the model received the input, whereas levels 1 and 2 are responsible for the recognition of the local and global patterns of auditory sequences in each experimental block. Dynamics of prediction and prediction error signals at each hierarchical model level are calculated based on the predefined (for each block) sequence structure (number of tones) and occurrence (distribution of xx, xY and xo sequences). This information is used to extract respective neural signatures corresponding to the prediction and prediction errors. With this setup, the authors show that the brain use different frequencies to propagate information up and down in the hierarchy. Specifically, the gamma-band is associated with the bottom-up prediction errors, whereas the beta-band relates to the top-down predictions. These results nicely complement a number of studies that suggest frequency specialization for the bottom-up (gamma) and top-down (beta) information flows. The results further suggest that predictions about local and global regularities in sensory stimuli are propagated with different beta-bands: higher and lower frequency ranges respectively. That is the notion of “frequency ordering” in hierarchically organized stimuli, “requires” nested predictions. The study provides an interesting approach that can be useful for identifying interactions between multiple processing hierarchies. If the framework could be extended to other modalities or scaled up for more complex sensory signals, it could become an important tool to understand hierarchical processing in the brain.

We thank the reviewer for understanding the key point of our paper.

Although the fitting results are overall consistent with existing neurophysiological evidence for predictive coding, the manuscript could be improved by providing additional information to enhance the biophysical plausibility of the model. In particular:

- *The model relies on specific hypotheses about the form of prediction and prediction error signals, i.e. the prediction error neuron performs a subtractive operation, whereas the prediction neuron receives no direct input but outputs the same prediction signal once in the steady state.*

We agree that our model needs more explanation. There are predictive coding models with considerations of biophysical plausibility. For example, the hierarchical architecture for empirical bayes (Friston K, *Phil. Trans. R. Soc. B*, 2005), asymmetric hierarchical message-passing (Arnal LH and Giraud AL, *Trends in Cognitive Sciences*, 2012), and corticocortical networks for predictive coding (Urgen BA and Miller LE, *J Neuroscience*, 2015). Despite different scales and levels of details, they all share the same core architecture, where each hierarchical level sends down a prediction signal and sends up a prediction-error signal after a subtractive operation. Our proposed model simply captures this feature (see figure below), without considering the details of how prediction and prediction-error signals interact within each hierarchical level. Thus, our illustration in the original submission, which shows that the prediction neuron receives no direct input, is unnecessary and confusing. To clarify this in the revision, we change the illustration in Figure 2 (also Figure 8C) and the corresponding texts.

While this might apply for the particular paradigm used in the study, I would hesitate to generalize it to another setting, where e.g. one could also consider spike patterns, etc. Could the authors refer to existing literature to provide rationale of these very specific hypotheses?

In addition to the universal predictive coding architecture, our model assumes that the strengths of predictions and prediction errors are encoded in neural firing rates. In the Results, we clarify this with a new reference (Pouget et al, *Nature Neuroscience*, 2013): “Absolute values are taken because we assume predictions and prediction errors are encoded in neuronal firing rates, a most straightforward scheme for encoding probabilistic representations and computations⁴⁰, which can only have non-negative values.” (page 8, lines 6-8)

Besides the rate coding, another potential mechanism of how neuronal populations can represent probabilities is based on the idea of “basis functions”, which is related to “spike patterns” mentioned by the reviewer. Based on the theory, different neurons or neuronal populations represent different probability distributions (i.e. basis functions), and their firing patterns determine how the corresponding basis functions are combined to represent the target probability distribution. However, the combination of basis functions is still weighted by the firing rate. We briefly discuss this in the Discussion: “To understand the dynamic process of prediction updating and error minimization, it is essential to examine how probabilities are encoded. It is thought that probability distributions, or their log values, are encoded straightforwardly in population firing rates (as adopted in our model), combinational firing patterns of neuronal populations representing specific probability distributions (called basis functions), or the value of membrane potentials⁴⁰, and their updates based on prediction errors are mediated by neuromodulators, such as acetylcholine⁷⁸.” (page 28, line 20 to page 29, line 1)

- What is the neurophysiological basis of the construction of separate X and Y streams, and for the form of input (1 for the target, 0 for anything else)?

The two-stream design is based on the tonotopic organization in the primary auditory cortex, where stimuli with distinct frequencies are processed and represented by different neuronal populations. Therefore, each stream can only receive sensory inputs of the corresponding

target frequency. However, we acknowledge that these populations need to interact in order to compute transition and sequence probabilities. We clarify this in the new model illustration in Figure 2, and in the Results:

“Furthermore, even though the x and y tones are processed in separate streams based on the tonotopic organization, two streams need to integrate information at Levels 1 and 2 to compute transition probabilities (TP_x , TP_y , and TP_o) and sequence probabilities (SP_{xx} , SP_{xy} , and SP_{xo}), respectively. In Figure 2C, we indicate these integrations for probability computations as horizontal gray bars between populations x_1 and y_1 and between populations x_2 and y_2 .” (page 9, lines 5-10)

Since the model fitting indirectly relies on the contrast between different conditions, fundamentally different model configurations, such as without predictive coding or predictive routing (Bastos et al. 2020, PNAS), could presumably reach similar fitting results. Therefore, it would be important to either provide solid physiological evidence for the basic hypotheses or to compare between radically different hypotheses (not just slight variation of the current model).

We agree. In the revision, we introduce alternative models to fit the same EEG data and compare their performance. In contrast to the proposed model, a two-level predictive coding model based on both transition and sequence probabilities (denoted by *2-level PC:TP+SP*), we tested three alternative models: (1) a single-level predictive coding model with only transition probability (*1-level PC:TP*), (2) a single-level predictive coding model with only sequence probability (*1-level PC:SP*), and (3) an adaptation-only model with no predictive coding mechanisms (*Adaptation-only*). We also used the model-free data-driven decomposition (*Model-free*), which provided the optimal and unbiased description of the data, as a benchmark.

For model comparison, we quantified the goodness of fit by using the Bayesian information criterion (BIC) and Akaike information criterion (AIC), which penalize models with more variables. Their calculations are summarized in a new Table 1.

Below are the items related to the model comparison:

- A new section in the Results titled “Model Comparison: Alternative Predictive Coding and Adaptation-Only Models” (page 15, line 13 to page 18, line 5)
- A new section in the Methods titled “Model comparison” (page 37, lines 16-24)
- A new Table 1
- New Figures 4C to 4G
- New Supplemental Figures 5, 7, and 8

We couldn't model predictive routing (Bastos et al. PNAS, 2020), since it's a descriptive model without a clear quantitative definition on how feedback and feedforward signals are modulated by sensory predictability.

Some detailed comment/questions:

-Line 7-10, page 3: I would suggest to also indicate that the predictions are updated to minimize prediction errors. Otherwise, it reads as prediction and prediction errors are “just” generated without any goal.

We address this in the revision:

“In a highly recursive process, higher-level cortical areas harboring internal models of the world predict inputs from lower-level areas through top-down connections, and then prediction-error signals are generated through bottom-up connections to update the internal models in order to minimize prediction errors.” (page 3, lines 7-10)

-Line 12-19, page 4: the authors mentioned a few examples of the correlation between (changes in) beta/alpha band activity and (changes in) prediction signals. How could the result of the current model provide mode specific explanations to such correlations?

In our paper, the high-beta P1 signal also correlated to predictability (transition probabilities), which can be described by Equations 3 and 4:

$$P1_x = s_0^{n-1} * TP_x \quad [3]$$

$$P1_y = TP_y \quad [4]$$

However, the correlation between P2 and predictability is more complicated, which is determined by both the sequence and transition probabilities. This can be described by Equations 7 and 8:

$$P2_x = SP_{xx} * (|s_0^{n-1} - P1_x * s_1|) + (1 - SP_{xx}) * P1_x * s_1 \quad [7]$$

$$P2_y = SP_{xy} * (|1 - P1_y * s_1|) + (1 - SP_{xy}) * P1_y * s_1 \quad [8]$$

where $P1_x$ and $P1_y$ can be replaced by transition probabilities as in Equations 3 and 4.

None of the studies on prediction signals mentioned in the Introduction investigated their hierarchical organization, which is a fundamental element of predictive coding theory that is often overlooked.

We clarify our goal and strategy in the Introduction, with the emphasis of this hierarchy aspect:

“However, how prediction and prediction-error signals interact across functional hierarchies, another fundamental element of predictive coding theory, remains unknown. To identify hierarchical prediction and prediction-error signals, we provided a quantitative definition of these signals based on a mechanistic and hierarchical predictive coding model, where predictions at each hierarchical level are generated to minimize the mean-squared prediction errors received at the same level. This allows us to infer the interactions between prediction and prediction-error signals within and across hierarchies when prediction is manipulated.” (page 5, lines 5-12)

-L8-22, P6: it is not clear that each block contains either 2-tone or 3-tone sequences (and not mixed 2 and 3 tone sequences). Although one could conclude it from e.g. diagram in Fig 1, mentioning it in the main text would avoid confusion about whether xxo (3-tone with omission) and xx (2-tone) sequences can occur within the same block and whether they are differently treated during analysis.

We clarify this in the Results:

“Note that xo contained only one stimulus items, but was used to represent an omission in a 2-tone sequence. Similarly, xxo was used to represent an omission in a 3-tone sequence. Sequences were delivered in blocks of 144 trials, which consisted of either only 2-tone sequences (xx , xy , and xo) or only 3-tone sequences (xxx , xyx , and xxo). Eight blocks were used, each with a distinct configuration of the sequence length and trial numbers for xx , xy , and xo (Figure 1B).” (page 6, lines 13-18)

Furthermore, in the Methods section, it is mentioned that each block was delivered twice, one time with tone A and the other time with tone B. However, in the main text it is not clear whether the authors treat predictions for tone A and tone B differently (e.g. P1x for tone A, is the same as P1x for tone B? (same goes for P2x, P1Y, P2Y)).

We clarify this in the Results:

“To examine the brain responses influenced by these probabilities, we eliminated tone-specific effects by delivering each block twice (one run with a low-pitched tone A as x and a high-pitched B as y, and the other run with tone B as x and tone A as y), and merged the EEG data from two runs for analysis (see Methods).” (page 7, lines 3-6)

This merging process took place when we calculated ERSP. In the Methods section, we clarify:

“For each trial type, the ERSP was calculated by averaging the normalized TFRs from the corresponding trials including both block A and block B to eliminate tone-specific effects.” (page 35, lines 16-18)

- L13, P9: it would be useful to provide discussion about how the minimization of mean-square error might be biophysically implemented.

It remains unclear how prediction errors are minimized (in the form of mean-square error or others). One candidate theory in neuromodulation, particularly acetylcholine. We discuss this in the Discussion:

“To understand the dynamic process of prediction updating and error minimization, it is essential to examine how probabilities are encoded. It is thought that probability distributions, or their log values, are encoded straightforwardly in population firing rates (as adopted in our model), combinational firing patterns of neuronal populations representing specific probability distributions (called basis functions), or the value of membrane potentials⁴⁰, and their updates based on prediction errors are mediated by neuromodulators, such as acetylcholine⁷⁸.” (page 28, line 20 to page 29, line 1)

-L15, P10: although the authors showed later that the model fitting is better when an additional x-o transition is considered at the end of an xx sequence, such construction changes the expected length of the particular sequence hence the temporal prediction. If this is considered in the contrast response, the model fit may be affected.

First, our interpretation of the inclusion of x-o transition at the end of an xx sequence is that the local prediction simply operates as Markov chains between three states: x, y, and o. That is, when receiving an x tone, P1 simply predicts what's coming next, without caring whether it's the first, second, or last tone, which is the information handled by P2.

Nonetheless, as pointed out by the reviewer, the additional prediction signal that anticipates another x tone at the end of an xx sequence changes the expected length and can cause an additional omission error. We believe that this additional omission error cannot be avoided at the local level, since a sequence ending cannot be expected by the local prediction, and will need to be managed at the global level. This was discussed in the Discussion and indicated in Figure 8D, where step 9 represents the additional error at the local level and step 12 represents the error at the global level that is reduced by P2. We discuss this in the revised Discussion:

“Furthermore, the error signals at steps 9 and 12 only occur in the xx sequence, since they are caused by the additional local prediction from the last x tone in the xx sequence. Therefore, in the within-block contrasts (xy – xx or xo – xx), and they could underlie the negative gamma oscillations observed at longer latencies (see Figures 5B and 5E).” (page 24, line 25 to page 25, line 4)

-L3, P12 and Fig 3B: why are there still negative values in the POS condition and positive values in the NEG condition? It is unclear to me what is plotted in these figures, z-scores?

In POS+NEG, POS, or NEG, all signal values (P1, P2, PE1, and PE2) are positive. But their contrast values between trial types (as shown in Figure 3) can be negative. For example, PE1 could be 0.2 and 0.3 in the xy and xx sequences, respectively (both positive), but their contrast xy–xx is –0.1 (a negative value). We clarify this in the Results:

“Note that while both PE1 and PE2 are positive values, their contrast values between two trial types can be negative.” (page 11, lines 17-18)

Also, for all three figure panels, PE1 and PE2 are very similar. How should we interpret such results?

This is because for each trial type (before the contrast), bigger PE1 usually leads to bigger PE2 (since PE1 cannot be fully explained by P2). This trend will remain after the contrasts (shown in Figure 3B). This similarity will affect the data decomposition performance. The perfect scenario will be that they are independent (orthogonal), but this condition is impossible to establish by any local-global designs (we have tried). However, data decomposition is also greatly influenced by the data itself (not only the model prediction shown as the third dimension in Figures 5 and 6).

-L15, P14: it seems that the text should have pointed to Figure 3B (and not Figure 2B).

Corrected. (page 13, line 19)

-L7-8, P15, Fig 4E-F: what are the functional interpretations of these areas? PE1 shows a slightly higher activation in the left compared to the right temporal area. Does this indicate auditory processing?

The topographical distribution shown in Figures 5A (originally Figure 4E) is based on reference-free current source density (CSD) transformation, which has been shown to be effective in highlighting the contribution of the underlying cortical sources spatially close to the electrodes (e.g., Kayser and Tenke 2006; Tenke and Kayser 2005, 2012). We address this in the Results with new references:

“Spatially, PE1 represented a source of bilateral auditory cortices, as evidenced by similar CSD-based distribution linked to auditory processing in other human studies^{43–45}. On the other hand, PE2 distribution represented a source of frontal cortex, as evidenced by similar CSD-based distribution linked to the medial prefrontal cortex or dorsal anterior cingulate cortex^{46–48}.” (page 17, lines 19-23)

-L20-21, P16: as the authors pointed out here, PE1 and PE2 are always highly correlated even with a lower adaptation factor (is it true that the similarity between PE1 and PE2 is correlated with the adaptation factor?). This confound does not seem to be considered in the analysis of within-block contrast that separated PE1 and PE2.

Yes, the similarity between PE1 and PE2 changes under different adaption factors, and this is why the same data will have different decompositions under different adaption factors.

As mentioned in a previous response, data decomposition is also greatly influenced by the data itself. For example, unstructured data cannot be decomposed well even PE1 and PE2 are independent (orthogonal). This is why we first used an unbiased data-driven approach to determine how many structured latent components in the data. Based on the consistency of the data-fitting for across-block contrasts (Figure 4E), there were only 3 structured components in the data instead of 4, and that's why we hypothesized that the high similarity between PE1 and PE2 collapsed the two components.

If we forced to fit the data with 4 components (P1, P2, PE1, and PE2, as shown in Figure 3C), we got a low fitting consistency and invalid decomposition results. The invalid results are shown below, where frequency-specific beta-band components in P1 and P2 were somehow preserved, but the gamma-band components in PE1 and PE2 were not separated. Note that we show this results here simply for demonstration, not for further interpretation.

*-L23, P16: why only consider equal contribution of PE1 and PE2? Fitting PE1 + p*PE2 might provide different results.*

We appreciate the suggestion, and add this in the Results:

“Therefore, for the model-driven analysis, we factorized the total contrast responses with the third dimension fixed with the predicted values of P1, P2, and $a*PE1+(1-a)*PE2$ from the model, where a was a weighting factor between 0 and 1. The best-fitting model with the smallest RSS was found with a consistency of 85% when $s_0 = 0.3$, $s_1 = 0.8$, $s_2 = 1.0$ (Figure 4F), and when $a = 0.5$ (see Supplementary Figure 3). This suggested that the identified components subserved P1, P2, and overall prediction errors (PE1+PE2).” (page 14, lines 6-11)

The fitting results are shown in a new Supplementary Figure 3.

L20-25, P17: Authors indicate that predictions of faster and slower events are encoded in fast frequency (~23 Hz beta) and slow (~15 Hz beta) bands. These results are derived from averaged neural signatures of predictions from 2-tone and 3-tone sequences. It would be interesting to see what frequency bands would appear for fast and slow events if one only considers either 2-tone or 3-tone sequences. Would the frequency ratio of slow/fast predictions follow the temporal regularity of slow-fast events in the stimuli (e.g., the difference between frequency bands to encode fast/slow events would be higher for 3-tone sequences)?

We fully agree that data-fitting on 2-tone and 3-tone sequences separately will generate more insights, not only on the frequency ratio as suggested by the reviewer, but also the timing when the second-level prediction signal (P2) is evoked (e.g. whether it's evoked by the first tone as we proposed in Figure 8D).

We tried, but unfortunately, models with 2-tone only or 3-tone only didn't yield high consistency in across-block contrasts. This is because P1 and P2 in the contrasts across 2-tone blocks (Contrast 1~12 in Figure 3C as shown below) are highly and negatively correlated, while P1 and P2 in the contrasts across 3-tone blocks (Contrast 13~24) are highly and positively correlated. As long as P1 and P2 are highly correlated (negatively or positively), it will be difficult to separate them, which was proven by the low fitting consistencies in our early analysis (data not shown). However, by combining them together, P1 and P2 became close to orthogonal (first 12 contrasts shared opposite signs and the last 12 contrasts shared the same sign), and decomposition became consistent. This is why we used both sequences to extract P1 and P2 in the across-block contrasts.

C

-L20-22, P18: As subjects learn x , they also learn about the violations, which may result in smaller surprise for xY in steady states. It might be useful to fit different models to the learning vs. the steady phase and compare the parameters.

Our results here are consistent with the Bayesian updating (data not shown, we removed this modeling part from the paper due to the length). Nonetheless, the reviewer's theory could be true too, since Bayesian updating is not the only learning mechanism. To test this theory, a learning model is required. However, to our knowledge, the only existing model that provide a quantitative definition of prediction and prediction-error signals during trial-by-trial learning is the Hierarchical Gaussian Filter (HGF). However, it's implementation in hierarchical prediction is limited (despite the term "hierarchical" in the name, which refers to a motor part of the model).

We discuss this issue in the "Prediction Update" section in the Discussion:

"One candidate to incorporate these ideas is a Bayesian model called the Hierarchical Gaussian Filter⁷⁹, which updates predictions by precision-weighted prediction errors^{1,3,12} and was implemented to examine prediction-error signals during learning in the brain^{73,80-83}. However, it's implementation in hierarchical prediction is limited (despite the term "hierarchical" in the name, which refers to a motor part of the model) but highly demanded." (page 29, lines 1-6)

-L21-22, P22: temporally unpredictable, but the tone identity is still predictable, therefore it might not be the case of "no prediction".

We agree that the identity of the first tone is predictable. That is, if the first tone suddenly becomes y , it will likely lead to a bigger surprise (compared to the case when the first tone is x). However, this prediction will not be carried by the theorized P1 and P2, which encodes the tone-to-tone transition and sequence identify (in this case is determined by the last tone), respectively. Therefore, there will be no P1 and P2 for the first tone, but there could be a different type of prediction as suggested by the reviewer.

Additional general questions:

-Why was an adaptation factor applied to the prediction error signal, but not to the prediction signal?

We add two additional scaling factors to P1 and P2, as described in the Results:

"At Levels 1 and 2, we added scaling factors s_1 and s_2 to the first-level predictions ($P1_x$ and $P1_y$) and the second-level predictions ($P2_x$ and $P2_y$), respectively, to account for imperfect predictions. When $s_1=1$ and $s_2=1$, the predictions are optimal (see how the optimal predictions were quantified below). When $s_1<1$ or $s_2<1$, the predictions are hypo-sensitive to the inputs. For example, if $s_1=0$, there will be no first level prediction. When $s_1>1$ or $s_2>1$, the predictions are hyper-sensitive to the inputs, where the corresponding transition or sequence probabilities are overestimated. Note that s_1 and s_2 were applied to both the x and y streams, since erroneous estimation of transition or sequence probabilities could occur at both streams." (page 9, line 18 to page 10, line 2)

Below are the new items related to this new analysis:

- Revised Figure 2D: model explanation
- New Figures 4D and 4F: optimal parameters

- New Supplemental Figures 4, 5, and 7

-How would information encoded in Level 2 be read out from a presumably higher-level neuron to infer the input sequence?

One straightforward possibility is multi-sequence modules, where the order of multi-tone sequences is structured. But it can be any structure with either a longer timescale or greater abstraction. We briefly discuss this at the end of the Discussion:

“These results advance the physiological measurement and modeling of predictive coding, and provide a platform to examine predictive signaling beyond two hierarchical levels (e.g. information of longer timescales or greater abstraction) and among multiple sensory modalities in normal and disordered brain.” (page 29, lines 20-23)

In addition, a rather “cosmetic” suggestion would be to re-evaluate notations and naming of sequences. The authors did a formidable job including all sequence information in the names (e.g. xY22), but a bit more consistency (why Y is with a capital letter and x not: then, it gets confusing later on, with capital letters of PE and P, TP, etc in math equations) and use of superscripts and subscripts (e.g. superscript to indicate the model level and subscript to indicate sequence type, thus P(1)(x) instead of P1x) would increase the readability of the text.

We fully agree. We change the followings in the revision to improve the readability.

- We simplify the block name by changing name such as “xY22” to “Block 5” (see revised Figure 1B). This is because that in the revision, the key points of the 8 blocks are simply the transition and sequence probabilities they created (as shown in the revised Figure 1B), and the information intended to show in the notation xY22 is no longer necessary.
- We also clear up the notations throughout the paper. The changes are listed below:

Old	New
Y (capital letter)	y (small letter)
TP _x , TP _Y , TP _o	TP _x , TP _y , TP _o
SP _{xx} , SP _{xY} , SP _{xo}	SP _{xx} , SP _{xy} , SP _{xo}
P1 _x , P2 _x , P1 _Y , P2 _Y	P1 _x , P2 _x , P1 _y , P2 _y
MSPE1 _x , MSPE2 _x	MSPE1 _x , MSPE2 _x
TN _x , TN _Y , TN _o	TN _x , TN _y , TN _o
r_shuffle	r_shuffle
a (adaptation factor)	s ₀ (also new variables s ₁ , s ₂ , and τ ₀)
A, B, x, Y, o (Tone/stimulus)	A, B, x, y, o (italic)

Reviewer #2 (Remarks to the Author):

Chao et al. have produced a remarkable advance, using a quantitative model to predict EEG responses responsible for prediction signals. The authors provide a quantitative definition of prediction error and prediction signals in the local/global paradigm with different manipulations of stimulus probabilities. This allows them to identify the spatio-spectral-temporal signatures of prediction. The work is excellent, technically sound, and provides a substantial advance. I have only one major concern and a few minor textual suggestions.

We thank the reviewer for understanding the key point of our paper.

Major Concern

Is the adaptation model by itself better or worse at explaining the EEG data than the PE1/PE2/P1/P2 model? In other words, could the authors quantify the amount of explained variance in their spatio-temporal-spectral data that is accounted for by adaptation alone, compared to the predictive coding model (and perhaps the predictive coding model considering only PE1 and P1)? The best way to make this comparison would be to use Bayesian model comparison, where the larger number of model parameters of the first and second level predictive coding models would penalize the model fit (forcing these models to explain more variance if they are to win as the “best” model). If the authors would be able to show this, it would show that a model consisting of top-down and bottom-up factors explains the data better than adaptation alone.

We fully agree. In the revision, we introduce alternative models to fit the same EEG data and compare their performance. In addition to the proposed model, a two-level predictive coding model based on both transition and sequence probabilities (denoted by *2-level PC:TP+SP*), we tested three alternative models: (1) a single-level predictive coding model with only transition probability (*1-level PC:TP*), (2) a single-level predictive coding model with only sequence probability (*1-level PC:SP*), and (3) an adaptation-only model with no predictive coding mechanisms (*Adaptation-only*). We also used the model-free data-driven decomposition (*Model-free*), which provided the optimal and unbiased description of the data, as a benchmark.

For model comparison, we quantified the goodness of fit by using the Bayesian information criterion (BIC) and Akaike information criterion (AIC), which penalize models with more variables. Their calculations are summarized in a new Table 1.

Below are the items related to the model comparison:

- A new section in the Results titled “Model Comparison: Alternative Predictive Coding and Adaptation-Only Models” (page 15, line 13 to page 18, line 5)
- A new section in the Methods titled “Model comparison” (page 37, lines 16-24)
- A new Table 1
- New Figures 4C to 4G
- New Supplemental Figures 5, 7, and 8

Minor Concerns

The authors state in the introduction that previous studies have not mapped prediction signals, although as later acknowledged in the discussion, there is previous work. In particular, human fMRI (Summerfield et al., 2006, reference 7) have shown frontal to posterior connectivity consistent with prediction signals, and pre-stimulus activity was found

in monkeys by Bastos et al., 2020 (reference 32 in the manuscript). One missing evidence for neural encoding of prediction in spiking activity is the work of Bell, Ungerleider, et al., Current Biology, 2016. However, it is true that the literature has focused mostly on prediction error signals and the prediction signals themselves have been under-studied. Please revise the introductory text to acknowledge this. It does not take anything away from the present work, which is still quite an advance, because it shows quantitatively how different types of prediction are encoded in the spatio-temporal-spectral domain.

We add the new reference, and acknowledge that both Summerfield et al., 2006 and Bell, Ungerleider, et al., Current Biology, 2016 aimed to separate prediction and prediction-error signals. We re-structure the Introduction to clarify our goal, particularly, we address the reviewer's suggestion a new paragraph:

“To disentangle prediction and prediction-error signals, a dynamic causal model has been used to identify top-down functional connectivity that encoded predicted stimuli during a discrimination task when the stimulus predictability was manipulated⁷. In a similar task, a regression model was used to evaluate the latent contributions of predictions and prediction errors in spiking activity³⁷. However, how prediction and prediction-error signals interact across functional hierarchies, another fundamental element of predictive coding theory, remains unknown. To identify hierarchical prediction and prediction-error signals, we provided a quantitative definition of these signals based on a mechanistic and hierarchical predictive coding model, where predictions at each hierarchical level are generated to minimize the mean-squared prediction errors received at the same level. This allows us to infer the interactions between prediction and prediction-error signals within and across hierarchies when prediction is manipulated. With this computational strategy, we recorded human EEG data during an auditory local-global paradigm with manipulated stimulus predictabilities at two hierarchies, and used a model-fitting approach to extract prediction and prediction-error signals from the EEG responses by a tensor-based decomposition method^{20,38,39}, and revealed their spatio-spectro-temporal structures and hierarchical interactions” (page 5, lines 1-16)

Page 24, line 24, “velocity information” is used in an awkward/non-standard way, difficult to parse. Please re-phrase.

We correct it:

“However, empirical evidence of how the timings of predictions are tuned across hierarchies is lacking.” (page 25, line 25 to page 26, line 1)

Page 25, lines 16-18, in referencing gamma/beta to superficial/deep layers, the authors do not cite any empirical papers. The appropriate reference here is Buffalo, Fries, Desimone et al., 2011 PNAS.

We add the reference. (page 26, line 6)

Page 25, lines 18-21, this distinction in terms of prediction errors using faster frequencies than the predictions that generate them was also a theoretical feature of the predictive coding model put forth by Bastos et al. (reference 42).

We add the reference. (page 26, line 22)

Could the authors make the code for determining the PE1/PE2/P1/P2 signals in the

local/global oddball paradigm publicly available for download? That would help other authors to be able to also use this quantitative framework and would greatly contribute to advancing the field.

Yes. We provide a MATLAB code for the model calculation.

Reviewer #3 (Remarks to the Author):

Reviewer #3 (Remarks to the Author):

Chao et al. measure brain responses while presenting auditory stimulus sequences, with the aim of finding neural correlates of predictive coding. They are specifically targeting correlates indicating prediction (as opposed to correlates of prediction error transmission, which have been reported in numerous studies). By a combination of tensor decomposition analysis and neural modeling, they report having found hierarchical prediction signals in the human brain. These prediction signals are reported to be "frequency-ordered" in the sense that the high beta frequency band represents tone-to-tone transition, and low beta band represents multi-tone sequence structure (and then gamma band reflects prediction error).

We thank the reviewer for understanding the key point of our paper.

Researchers have tried to find correlates of prediction and dissociate these from prediction error signals for many years. The key towards achieving this is to use a paradigm that allows for a clear distinction between prediction and prediction error. Unfortunately, the paradigm used in the current study does not fulfill this requirement in a more convincing manner than previous studies have done.

Respectfully, we feel the reviewer misidentified the critical logic of our study. We do not claim that the local-global paradigm allows the separation of prediction and prediction error at the neurophysiological level. Rather, our motivation for using the local-global paradigm is to investigate the computational representations of these two signals across functional hierarchies, a fundamental element of predictive coding theory that is often overlooked.

Although we focus on the predictive hierarchy, we believe that our study also advances the debate on separation of prediction and error signals. Previous studies on prediction signals manipulated the predictability of sensory inputs, and compared the neural responses under different conditions. However, manipulating predictability changes not only the prediction signal, but also the subsequent prediction-error signal. Therefore, we agree with the reviewer that all current paradigms cannot disentangle prediction error from prediction. Another approach is to examine the neural responses during omission based on the argument that when the sensory input is omitted what's left is the prediction signal. However, this method is also inaccurate, since omissions also lead to surprises or omission errors.

Our study goes beyond the published literature because we assume as a starting point that prediction and prediction-error signals are dependent on each other within and across hierarchies. The key insight in our study is that we provide a clear quantitative definition of prediction and prediction-error signals, allowing us to infer their hierarchical interactions and interdependence when prediction is manipulated. With this computational strategy, we created a quantitative model based on the predictive-coding framework and 8 experimental conditions with combinations of local and global predictability, which enabled us to use model-fitting to extract prediction and error signals from the neural responses.

The utility of our computational strategy and its value for the field is that it can be applied to any experimental paradigm where predictability can be defined, not just the local-global paradigm. Again, the reason we targeted the local-global paradigm is for its hierarchical design.

We address these points in the last 3 paragraphs in the Introduction (pages 4-5). We also clarify our goal in the revised Abstract:

“However, the identification of feedback prediction signals has been elusive due to their causal entanglement with prediction-error signals. Here, we used a quantitative model to decompose these signals in electroencephalography during an auditory behavior, and identified their spatio-spectral-temporal signatures across two functional hierarchies.” (page 2, lines 4-8)

In fact, the authors do not explain the underlying logic of their paradigm. Instead they create the impression that they use a well-validated “local-global paradigm”, with reference to Bekinschtein et al. (2009) and to their own prior work in primates (Chao et al., 2018, Neuron). The scientific strength of this local-global paradigm is that local predictions in 5-tone sequences are played against global predictions in terms of expecting A or B as the final tone of a sequence starting with AAAA.. In the current study, no 5-tone sequences are employed but only 2-tone sequences. Hence by all common knowledge on how the brain extracts auditory patterns, there will be no local prediction in the second position. The allusion to the local-global paradigm is thus misleading. This would be only an issue of framing if the authors’ paradigm was convincing in and of itself, but it is really not obvious how their paradigm solves the dissociation of prediction and prediction error.

The reviewer’s assertion that there will be no local prediction in the second position in 2-tone sequences is mistaken. Local prediction at the second position is established by transitions from the first to the second items ($x \rightarrow x$, $x \rightarrow Y$, or $x \rightarrow o$), which are estimated by the transition probabilities (conditional probabilities $p(x|x)$, $p(Y|x)$, or $p(o|x)$). Importantly, the transition probabilities are different from sequence probabilities, since transition probabilities include the omission at the end of the xx sequence (see the revised Figure 1B below, showing distinct combinations of transition and sequence probabilities in our block design). Prediction of the last tone in a 2-tone sequence is similar to predicting the last tone in a 3- or 5-tone sequence, which are based on distinct transition and sequence probabilities regardless of sequence length.

The reviewer’s statement that we used only 2-tone sequences is false. We not only used 2-tone sequences but also 3-tone sequences (please see Figure 1A below). The key difference between 3-tone and the 5-tone sequences is the transition probability, and we intentionally selected 2-tone and 3-tone instead of 5-tone sequences, since the transition probability $p(x|x)$ for the 5-tone sequence is too high (close to 1) and therefore the contrast is too small to detect. Indeed, our local-global paradigm is specifically designed for its superior decomposition.

On the contrary, there are some problematic assumptions in the paradigm, such as that there can be only neuronal adaptation for tones that appear twice in a sequence (ignoring the fact that adaptation can take place across sequences, i.e. spanning longer timescales).

We agree with the reviewer’s concerns and have addressed them with a new analysis. In the revision, we tuned the model not only at the sensory level, but also the prediction at the first and second levels. Specifically, we added scaling factors to the sensory input (s_0 , originally “a”), $P1$ (s_1) and $P2$ (s_2), and we explored different combinations of s_0 , s_1 , and s_2 . This is described in the Results:

“At Levels 1 and 2, we added scaling factors s_1 and s_2 to the first-level predictions (PI_x and PI_y) and the second-level predictions ($P2_x$ and $P2_y$), respectively, to account for imperfect predictions. When $s_1 = 1$ and $s_2 = 1$, the predictions are optimal (see how the optimal predictions were quantified below). When $s_1 < 1$ or $s_2 < 1$, the predictions are hypo-sensitive to the inputs. For example, if $s_1 = 0$, there will be no first level prediction. When $s_1 > 1$ or $s_2 > 1$, the predictions are hyper-sensitive to the inputs, where the corresponding transition or sequence probabilities are overestimated. Note that s_1 and s_2 were applied to both the x and y streams, since erroneous estimation of transition or sequence probabilities could occur at both streams.” (page 9, line 18 to page 10, line 2)

Below are the new items related to this new analysis:

- Revised Figure 2D: model explanation
- New Figures 4D and 4F: optimal parameters
- New Supplemental Figures 4, 5, and 7

Moreover, we also examined alternative models in the revision. We introduce alternative models to fit the same EEG data and compare their performance. In contrast to the proposed model, a two-level predictive coding model based on both transition and sequence probabilities (denoted by *2-level PC:TP+SP*), we tested three alternative models: (1) a single-level predictive coding model with only transition probability (*1-level PC:TP*), (2) a single-level predictive coding model with only sequence probability (*1-level PC:SP*), and (3) an adaptation-only model with no predictive coding mechanisms (*Adaptation-only*). We also used the model-free data-driven decomposition (*Model-free*), which provided the optimal and unbiased description of the data, as a benchmark.

For model comparison, we quantified the goodness of fit by using the Bayesian information criterion (BIC) and Akaike information criterion (AIC), which penalize models with more variables. Their calculations are summarized in a new Table 1.

Below are the items related to the model comparison:

- A new section in the Results titled “Model Comparison: Alternative Predictive Coding and Adaptation-Only Models” (page 15, line 13 to page 18, line 5)
- A new section in the Methods titled “Model comparison” (page 37, lines 16-24)
- A new Table 1
- New Figures 4C to 4G
- New Supplemental Figures 5, 7, and 8

Regarding the adaptation timescales as pointed out by the reviewer, we found that the best-fitting adaption-only model occurred when adaptation spans over multiple sequences: “We then fitted the EEG data with these predicted values, and the optimal parameters were found to be: $s_0 = 0.3$ and $\tau_0 \approx 1.4s$ (see Table 1 and Supplementary Figure S7). This indicated that in *Adaptation-only*, adaption with a timescale that covered multiple sequences was needed.” (page 16, lines 15-18)

The authors also implicitly assume that the brain only ever makes one prediction, whereas some studies suggest that two opposing predictions can be maintained in parallel.

This statement is incorrect. We did not implicitly assume that the brain makes only one prediction. In our model, there are two predictions at each level (e.g. P1x and P1Y), and they are in fact opposing. Overall prediction is maintained by 4 predictions at two hierarchical levels both in parallel and in series. Our model features realistic brain prediction complexity.

Furthermore, essential details about sequence construction are missing, such as whether rare events (eliciting prediction error) can immediately repeat or not. (If they can't, this changes predictability in the stimulus right after the prediction error.) It remains unclear which contrasts really indicate prediction.

The sequences were created with a pseudo-random approach. We added more details in the Methods to clarify this:

“The order of the sequences was pseudorandom within each block, where the total sequences were divided into four phases while each phase kept the same sequence ratios. For example, for Block 1, each phase had 24 trials of xx, 6 trials of xy, and 6 trials of xo (a total of 36 trials in a phase). The sequence order was randomized for each phase in each block, with possibilities of consecutive rare sequences (e.g. two consecutive xy sequences in Block 1). The reason for the pseudorandom order was to maintain overall sequence probabilities throughout the learning. Furthermore, the reason to allow consecutive rare sequences was to avoid introducing additional statistical structures into the sequences.” (page 32, line 21 to page 33, line 3)

In the current version, we explain how we extract prediction in the section “Prediction Signals Extracted From Across-Block Contrasts” in the Results. In the revision, clarify this in a new section in the Results titled “Model-Fitting: Optimal Decomposition of EEG Data” (page 12, line 2 to page 15, line 11).

One implicit assumption is that predictions must be learned across the block (nothing can be predicted at block start). This assumption would naturally lead to a modeling framework that specifically looks for a neural process that emerges during the course of the block (with sequence exposition). In contrast, the modeling approach identifies components that are present throughout the whole block, and then analyzes whether they change in amplitude between early and late portions (which gives a rather small effect, no striking evidence of emerging prediction). Hence altogether, although the study uses technically impressive approaches towards modeling, the underlying paradigm seems too weak to yield conclusive results.

We agree with the reviewer that the lack of trial-by-trial learning is a limitation of our model. There are published models that include learning dynamics, such as the Hierarchical Gaussian Filter (HGF) but its implementation in hierarchical prediction is limited. We discuss this issue in the “Prediction Update” section in the Discussion:

“One candidate to incorporate these ideas is a Bayesian model called the Hierarchical Gaussian Filter⁷⁹, which updates predictions by precision-weighted prediction errors^{1,3,12} and was implemented to examine prediction-error signals during learning in the brain^{73,80–83}. However, its implementation in hierarchical prediction is limited (despite the term “hierarchical” in the name, which refers to a motor part of the model) but highly demanded.” (page 29, lines 1-6)

Regarding the comment that our paradigm seems too weak to yield conclusive results, please see our replies on our strategy above (also addressed in the last 3 paragraphs in the revised Introduction). In brief, we do not claim that the local-global paradigm per se is conclusive with respect to separating prediction and error, but instead we assert that our model-fitting approach provides a valid route to studying hierarchical signals.

There are further, more minor details, such as inconsistencies in the description: "During the task, participants passively listened to a series of short tone sequences based on the local-global auditory paradigm while brain activity was recorded by 64-channel EEG. To ensure vigilance, participants were instructed to both visually fixate and attend to the sounds." Participants cannot "passively listen" and "attend" to the same sounds at once. This is a critical difference as it informs about the top-down contribution in acquiring predictions.

The term “passively” was used here to indicate that the participants were not required to produce behavioral outputs to react to the tone sequences. To avoid confusion, the term is removed. (page 6, line 3)

Since the paradigm is already tipped towards more global type of predictions (i.e., local stimulus-driven prediction contributions are weak, see above), heavy attention involvement would tip this balance even further and question the generalizability towards other predictive coding instantiations.

This criticism is based on an incorrect assumption. We have addressed the misunderstanding of weak local prediction in our paradigm above (where the revised Figure 1 is introduced). As pointed out by the review, it’s been shown that attention plays an important role in prediction, particularly the global prediction (Bekinschtein et al. *PNAS*, 2009; Chennu et al. *Journal of Neuroscience*, 2013). If attention affects the prediction (P1 and/or P2) in the task, we expect to see suboptimal s_1 and/or s_2 (e.g. $s_1 < 1$ and/or $s_2 < 1$). In fact, suboptimal s_1 and/or s_2 were found in a non-human primate model of autism (paper in preparation).

Reviewers' comments:

Reviewer #1 (Remarks to the Author):

Overall, I am happy with the manuscript revision and how the authors addressed my remarks. The authors did an laudable job and the manuscript has gained considerably in clarity.

Minor remarks:

-p3 lines: 7-10: there is also the notion of lateral prediction errors originating from connections within the same hierarchical level. This formulation seems that prediction errors are solely generated via bottom-up connections.

-p10 lines 10-12: unclear sentence.

-p16 lines 20-24: for model comparisons both BIC and AIC were used, which feels a bit redundant, maybe one of the techniques should be considered.

-Discussion: the manuscript can be further strengthened by discussing how the proposed network model could be biophysically (e.g. through a network of E-I neurons) instantiated and verified with finer-grained measurements such as LFP or single-neuron activities.

Reviewer #2 (Remarks to the Author):

I thank the authors for their attentiveness to detail and for their response. They have addressed my concerns.

However, there should be a mention in Table 1 and in the main text on the model comparison portion of the text, whether the differences in SSE/AIC/BIC values between the models was significant.

I congratulate the authors on an extremely innovative and important contribution to the field.

Reviewers' comments:

Reviewer #1 (Remarks to the Author):

Overall, I am happy with the manuscript revision and how the authors addressed my remarks. The authors did an laudable job and the manuscript has gained considerably in clarity.

Minor remarks:

-p3 lines: 7-10: there is also the notion of lateral prediction errors originating from connections within the same hierarchical level. This formulation seems that prediction errors are solely generated via bottom-up connections.

There are many proposals for the predictive coding architecture. Some include lateral signaling of predictions and prediction errors, some also contain self-recurrent connections. However, the common core components are the bidirectional connections between hierarchies. We modify the sentence and the sentence before to clarify this: “According to predictive coding theory, this form of dynamic communication is achieved by a hierarchical and bidirectional cascade of large-scale cortical signaling in order to minimize overall prediction errors. In a highly recursive process, higher-level cortical areas harboring internal models of the world predict inputs from lower-level areas through top-down connections, and prediction-error signals are generated to update the internal models through bottom-up connections.” (page 3, lines 5-11)

-p10 lines 10-12: unclear sentence.

We clarify the sentence:

“It is important to note that the transition from tone x to o occurs not only during the xo sequence, but also at the end of the xx sequence where the last x tone is followed by no tone. Since Level 1 simply predicts what will happen after an x tone and makes no distinction between the two cases, the x to o transitions in both the xo and xx sequences were considered in the transition probability calculation (see Methods).” (page 10, lines 10-15)

-p16 lines 20-24: for model comparisons both BIC and AIC were used, which feels a bit redundant, maybe one of the techniques should be considered.

We agree and remove AIC from the main text (page 16), Table 1, and Methods (page 37). We also rearrange the table (move the optimal parameters from the last to the second column) for clearer flow.

-Discussion: the manuscript can be further strengthened by discussing how the proposed network model could be biophysically (e.g. through a network of E-I neurons) instantiated and verified with finer-grained measurements such as LFP or single-neuron activities.

Our proposed model has two key features whose biological plausibility needs to be (and can be) tested. One is the positive and negative error computations, the other is the prediction computations that encode transition and sequence probabilities. We have discussed the former one based on the biologically plausible circuits proposed by Keller & Mrsic-Flogel (reference 71), and we add another candidate in the revision (New reference 76, Hertäg,

Loreen, and Henning Sprekeler. "Learning prediction error neurons in a canonical interneuron circuit." *Elife* 9 (2020): e57541):

"To identify the positive and negative error signals or test the theorized biological circuitries^{71,76}, one would need to use neural recordings with single-cell resolution, such as single-unit activity recordings or calcium imaging." (page 27 line 24 to page 28 line 1)

Regarding the prediction computations, which is a focal point of the paper, we add a paragraph in the revision under the section "Prediction Encoding and Updating" to explain how the proposed prediction mechanism can be implemented biologically:

"One important feature in our model is that while prediction is established in each individual stream, its value is determined by the stimulus probability (TP at Level 1 and SP at Level 2) which requires information integrated from both streams. Therefore, we believe that prediction is encoded in a neuronal network with four key features which can be tested by using finer-grained measurements such as single-unit activity recordings or calcium imaging: (1) inter-stream connections (spans spatially across streams), (2) probability encoding (changes activation based on sensory predictability), (3) proactive timing (activates before the sensory input), and (4) top-down regulation (influences responses at the lower hierarchy)." (page 29, lines 9-16)

Reviewer #2 (Remarks to the Author):

I thank the authors for their attentiveness to detail and for their response. They have addressed my concerns.

However, there should be a mention in Table 1 and in the main text on the model comparison portion of the text, whether the differences in SSE/AIC/BIC values between the models was significant.

I congratulate the authors on an extremely innovative and important contribution to the field.

We thank the reviewer for the critical comment. There are three approaches to assess the significance of the BIC values (in the second revision, we remove the results of AIC due to its redundancy, which is suggested by another reviewer).

#1. A theoretical approach. The difference in BICs between models (e.g. ΔBIC for Adaption-only – proposal model = $(-1.8115\text{e}+08) - (-1.8210\text{e}+08)$), which could reveal how much the proposed model is favored. When comparing models, a difference in BIC of 10 corresponds to the odds being 150:1 that the model with the more negative value is the better fitting model and is considered “very strong” evidence in favor of the model with the more negative BIC value (Raftery, Adrian E. "Bayesian model selection in social research." *Sociological methodology* (1995): 111-163). In our case, the differences in BIC between the proposed model and the alternative models were all greater than 10, thus the proposed model is strongly favored.

#2. An analytical approach based on model-fitting. We performed the model-fitting (via PARAFAC) with different initialization methods (e.g. random orthogonalized values), and expected to obtain different residual errors and thus different BIC values. However, this was not the case. Different initializations all led to the same result. This indicated the robustness of the model-fitting, but we did not observe the expected variability that can be used to test the significance of RSS and BIC.

#3. An analytical approach based on resampling. Another way to acquire variable RSS/BIC is to resample the data (e.g. leave one subject out each time), recalculate the contrast responses (1000 bootstrapping for each comparison), and quantify RSS/BIC. This approach will be very time consuming, considering minimally we need to repeat the same analysis in the paper (1000 bootstrapping for a total of 40 comparisons, and fit 10 models) at least 30 times (if we use the leave-one-subject-out approach). More importantly, we need to account for the variability in the model-fitting results from the 30 resamples, even though should be small, and modify the figures accordingly (e.g. Figures 5 and 6).

We decide not to pursue approach #3 because the value added will be limited while the expected time required for the drawn-out analysis and paper revision is extremely lengthy. Thus, we modify the main text and based on the approach #1 and add a new reference (reference 43):

“For model comparison, we quantified the goodness of fit by using the Bayesian information criterion (BIC), which penalizes models with more variables (see details in Table 1). For both the within- and across-block contrasts, our proposed model showed the best fitting with the BIC significantly lower than the alternative models (the between-model differences in BIC were greater than 10, which corresponded to a 150:1 odds that the proposed model was the

better fitting model ⁴³), and was close to the model-free data-driven results (Figure 4G, more details in Table 1).” (page 16 line 23 to page 17 line 4)